# DISTRIBUTED GRAPH NEURAL NETWORK TRAINING WITH PERIODIC STALE REPRESENTATION SYNCHRONIZATION

## ABSTRACT

Despite the recent success of Graph Neural Networks (GNNs), it remains challenging to train a GNN on large graphs with over millions of nodes & billions of edges, which are prevalent in many graph-based applications such as social networks, recommender systems, and knowledge graphs. Traditional sampling-based methods accelerate GNN training by dropping edges and nodes, which impairs the graph integrity and model performance. Differently, distributed GNN algorithms accelerate GNN training by utilizing multiple computing devices and can be classified into two types: "*partition-based*" methods enjoy low communication cost but suffer from information loss due to dropped edges, while "*propagation-based*" methods avoid information loss but suffer from prohibitive communication overhead caused by neighbor explosion. To jointly address these problems, this paper proposes DIGEST (DIstributed Graph reprEsentation SynchronizaTion), a novel distributed GNN training framework that synergizes the complementary strength of both categories of existing methods. We propose to allow each device utilize the stale representations of its neighbors in other subgraphs during subgraph parallel training. This way, out method preserves global graph information from neighbors to avoid information loss and reduce the communication cost. Therefore, DIGEST is both computation-efficient and communication-efficient as it does not need to frequently (re-)compute and transfer the massive representation data across the devices, due to neighbor explosion. DIGEST provides synchronous and asynchronous training manners for homogeneous and heterogeneous training environment, respectively. We proved that the approximation error induced by the staleness of the representations can be upper-bounded. More importantly, our convergence analysis demonstrates that DIGEST enjoys the state-of-the-art convergence rate. Extensive experimental evaluation on large, real-world graph datasets shows that DIGEST achieves up to $21.82\times$ speedup without compromising the performance compared to state-of-the-art distributed GNN training frameworks.

## 1 INTRODUCTION

*Graph Neural Networks* (GNNs) have shown impressive success in analyzing non-Euclidean graph data and have achieved promising results in various applications, including social networks, recommender systems and knowledge graphs, etc. (Dai et al., 2016; Ying et al., 2018; Eksombatchai et al., 2018; Lei et al., 2019; Zhu et al., 2019). Despite the great promise of GNNs, they meet significant challenges when being applied to large graphs, which are common in real world—the number of nodes of a large graph can be up to millions or even billions. For instance, Facebook social network graph contains over 2.9 billion users and over 400 billion friendship relations among users[1]. Amazon provides recommendations over 350 million items to 300 million users[2]. Further, natural language processing (NLP) tasks take advantage of knowledge graphs, such as Freebase (Chah, 2017) with over 1.9 billion triples. Training GNNs on large graphs is jointly challenged by the lack of inherent parallelism in the backpropagation optimization and heavy inter-dependencies among graph nodes, rendering existing parallel techniques inefficient. To tackle the unique challenges in GNN

---

[1] https://backlinko.com/facebook-users
[2] https://amzscout.net/blog/amazon-statistics

training, distributed GNN training is a promising open domain that has attracted fast-increasing attention in recent years. A classic and intuitive way is by *sampling*. Until now, a good number of graph-sampling-based GNN methods have been proposed, including neighbor-sampling-based methods (e.g., GraphSAGE (Hamilton et al., 2017), VR-GCN (Chen et al., 2018)) and subgraph-sampling-based methods (e.g., Cluster-GCN (Chiang et al., 2019), GraphSAINT (Zeng et al., 2019)). These methods enable a GNN model to be trained over large graphs on a single machine by sampling a subset of data during forward or backward propagation. While sampling operations reduce the size of data needed for computation, these methods suffer from degenerated performance due to unnecessary information loss. To walk around this drawback and also to leverage the increasingly powerful computing capability of modern hardware accelerators, recent solutions propose to train GNNs on a large number of CPU and GPU devices (Thorpe et al., 2021; Ramezani et al., 2021; Wan et al., 2022) and have become the *de facto* standard for fast and accurate training over large graphs.

Existing methods in distributed training for GNNs can be classified into two categories, namely "*partition-based*" and "*propagation-based*", by how they tackle the trade-off between *computation/communication cost* and *information loss*. "Partition-based" methods (Angerd et al., 2020; Jia et al., 2020; Ramezani et al., 2021) partition the graph into different subgraphs by dropping the edges across subgraphs. This way, the GNN training on a large graph is decomposed into many smaller training tasks, each trained in a siloed subgraphs in parallel, reducing communications among subgraphs, and thus, tasks, due to edge dropping. However, this will result in severe information loss due to the ignorance of the dependencies among nodes across subgraphs and cause performance degeneration. To alleviate information loss, "propagation-based" methods (Ma et al., 2019; Zhu et al., 2019; Zheng et al., 2020; Tripathy et al., 2020; Wan et al., 2022) do not ignore edges across different subgraphs with neighbor communications among subgraphs to satisfy GNN's neighbor aggregation. However, the number of neighbors involved in neighbor aggregation grows exponentially as the GNN goes deeper (i.e., *neighborhood explosion* (Hamilton et al., 2017)), hence inevitably suffering huge communication overhead and plagued training efficiency.

Therefore, although "partition-based" methods can parallelize a training job among partitioned subgraphs, they suffers from information loss and low accuracy. "Propagation-based" methods, on the other hand, use the entire graph for training without information loss but suffer from huge communication overhead and poor efficiency. Hence, it is highly imperative to develop a method that can jointly address the problems of high communication cost and severe information loss. Moreover, theoretical guarantees (e.g., on convergence, approximation error) are not well explored for distributed GNN training due to the joint sophistication of graph structure and neural network optimization.

To address the aforementioned challenges, we propose a novel distributed GNN training framework that synergizes the complementary strengths of both partitioning-based and propagating-based methods, named **DI**stributed **G**raph repr**E**sentation **S**ynchroniza**T**ion, or **DIGEST**. DIGEST does not completely discard node information from other subgraphs in order to avoid unnecessary information loss; DIGEST does not frequently update all the node information in order to minimize communication costs. Instead, DIGEST extends the idea of single-GPU-based GCN training with stale representations (Chen et al., 2018; Fey et al., 2021) to a distributed setting, by enabling each device to efficiently exchange a relatively stale version of the neighbor representations from other subgraphs, to achieve scalable and high-performance GNN training. This effectively avoids neighbor updating explosion and reduces communication costs across training devices. Considering naive synchronous distributed training that inherently lacks the capability of handling stragglers caused by training environment heterogeneity (e.g. GPU resource heterogeneity), we further design an asynchronous version of DIGEST (DIGEST-A), where each subgraph follows a non-blocking training manner. The synchronous version is a natural generalization of Fey et al. (2021) while the asynchronous version can handle the straggler issue (Chen et al., 2016; Zheng et al., 2017) in synchronous version and enjoys even better performance. From the system aspect, DIGEST (1) enables efficient, cross-device representation exchanging by using a shared-memory key-value storage (KVS) system, (2) supports both synchronous and asynchronous parameter updating, and (3) overlaps the computation (layer training) with I/Os (pushing/pulling representations to/from the KVS.

Furthermore, we proved that the approximation error induced by the staleness of representation can be bounded. More importantly, global convergence guarantee is provided, which demonstrates that DIGEST has the state-of-the-art convergence rate. Our main contributions can be summarized as:

- **Proposing a novel distributed GNN training framework that synergizes the benefits of partition-based and communication-based methods.** Existing work in distributed GNN training

focus on two *contradictory* objectives: partition-based methods target minimizing the communication cost while propagation-based methods aim to minimize information loss. DIGEST drops *no* edges while avoiding communication overhead by integrating the strengths of both categories.

- **Developing a periodic stale representation synchronization technique for distributed GNN training.** DIGEST utilizes the entire graph for training by separating in- and out-of-subgraph neighbor nodes and approximating the latter with stale representations. Instead of making strictly synchronous pull/push operations for the representations of all layers before/after training, DIGEST overlaps pull/push operations with layer training to minimize the overall training time. Furthermore, a shared-memory-based KVS is used among subgraphs for efficiently exchanging representations.

- **Providing extensive theoretical guarantee on both performance and convergence of the proposed algorithm.** We proved that DIGEST's convergence rate is $\mathcal{O}(T^{-2/3}M^{-1/3})$ with $T$ iterations and $M$ subgraphs, which is close to vanilla distributed GNN training without staleness. Convergence guarantee for both synchronous and asynchronous versions of DIGEST is provided. We also showed the upper bound on the approximation error of gradients due to the staleness.

- **Conducting comprehensive empirical results on both performance and speedup.** We perform extensive evaluation on four benchmark with classic GNNs (e.g., GCN (Hamilton et al., 2017) and GAT (Veličković et al., 2017)). The experimental results show that for the best case DIGEST improves the performance by **33.14**%, and achieves **21.82**× speedup in training time compared to two state-of-the-art distributed GNNs training frameworks.

## 2 BACKGROUND AND PROBLEM FORMULATION

In this section, we first introduce the Graph Neural Network (GNN) and its training on a single machine, and then formulate the problem of distributed GNN training.

**Graph Neural Networks.** GNNs aim to learn a function of signals/features on a graph $\mathcal{G}(\mathcal{V}, \mathcal{E})$ with node representations $\mathbf{X} \in \mathbb{R}^{|\mathcal{V}| \times d}$, where $d$ denotes the node feature dimension. For typical semi-supervised node classification tasks (Kipf & Welling, 2016), where each node $v \in \mathcal{V}$ is associated with a label $\mathbf{y}_v$, a $L$-layer GNN $f$ is trained to learn the node representation $\mathbf{h}_v$ such that $\mathbf{y}_v$ can be predicted accurately. The training process of a GNN can be practically described as the node representation learning based on the *message passing mechanism* (Gilmer et al., 2017). Analytically, given a graph $\mathcal{G}(\mathcal{V}, \mathcal{E})$ and a node $v \in \mathcal{V}$, the $(\ell + 1)$-th layer of the GNN is defined as

$$\mathbf{h}_v^{(\ell+1)} = f^{(\ell+1)}\Big(\mathbf{h}_v^{(\ell)}, \big\{\mathbf{h}_u^{(\ell)} : u \in \mathcal{N}(v)\big\}\Big) = \Psi^{(\ell+1)}\Big(\mathbf{h}_v^{(\ell)}, \Phi^{(\ell+1)}\big(\big\{\mathbf{h}_u^{(\ell)} : u \in \mathcal{N}(v)\big\}\big)\Big), \quad (1)$$

where $\mathbf{h}_v^{(\ell)}$ denotes the representation of node $v$ in the $\ell$-th layer, and $\mathbf{h}_v^{(0)}$ being initialized to $\mathbf{x}_v$ ($v$-th row in $\mathbf{X}$), and $\mathcal{N}(v)$ represents the set of direct 1-*hop* neighbors for node $v$. Each layer of the GNN, i.e. $f^{(\ell)}$, can be further decomposed into two components: 1) Aggregation function $\Phi^{(\ell)}$, which takes the nodes representations of node $v$'s neighbors as input, and output the aggregated neighborhood representation. 2) Updating function $\Psi^{(\ell)}$, which combines the representation of $v$ and the aggregated neighborhood representation to update the representation of node $v$ for the next layer. Both $\Phi^{(\ell)}$ and $\Psi^{(\ell)}$ can choose to use various functions in different types of GNNs. To train a GNN on a single machine, one can minimize the empirical loss $\mathcal{L}(\mathbf{W})$ over the entire graph in the training data, i.e., $\mathcal{L}(\mathbf{W}) = (1/|\mathcal{V}|) \sum_{v \in \mathcal{V}} Loss(\mathbf{h}_v^{(L)}, \mathbf{y}_v)$, where $Loss(\cdot, \cdot)$ denotes a loss function (e.g., cross entropy loss), and $\mathbf{h}_v^{(L)}$ denotes the representation of node $v$ from the last layer of the GNN and can be calculated by following Eq. 1 recursively.

**Distributed Training for GNNs.** Distributed GNN training means to first partition the original graph into multiple subgraphs without overlap, which can also be considered as mini batches. Then different mini-batches are trained in different devices in parallel. Here, Eq. 1 can be further reformulated as

$$\mathbf{h}_v^{(\ell+1)} = \Psi^{(\ell+1)}\Big(\mathbf{h}_v^{(\ell)}, \Phi^{(\ell+1)}\Big(\underbrace{\big\{\mathbf{h}_u^{(\ell)} : u \in \mathcal{N}(v) \cap \mathcal{S}(v)\big\}}_{\text{In-subgraph nodes}} \cup \underbrace{\big\{\mathbf{h}_u^{(\ell)} : u \in \mathcal{N}(v) \setminus \mathcal{S}(v)\big\}}_{\text{Out-of-subgraph nodes}}\Big)\Big), \quad (2)$$

where $\mathcal{S}(v)$ denotes the subgraph that node $v$ belongs to. In this paper, we consider the distributed training of GNNs with multiple local machines and a global server. The original input graph $\mathcal{G}$ is first partitioned into $M$ subgraphs, where each $\mathcal{G}_m(\mathcal{V}_m, \mathcal{E}_m)$ represents the subgraph $m$. Our goal is to find the optimal set of parameters $\mathbf{W}$ in a distributed manner by minimizing each local loss, i.e.,

$$\min_{\mathbf{W}} \mathcal{L}_m^{\text{Local}}(\mathbf{W}_m) = \frac{1}{|\mathcal{V}_m|} \sum_{v \in \mathcal{V}_m} Loss(\mathbf{h}_v^{(L)}, \mathbf{y}_v), \quad m = 1, 2, \cdots, M \ \text{in parallel}, \quad (3)$$

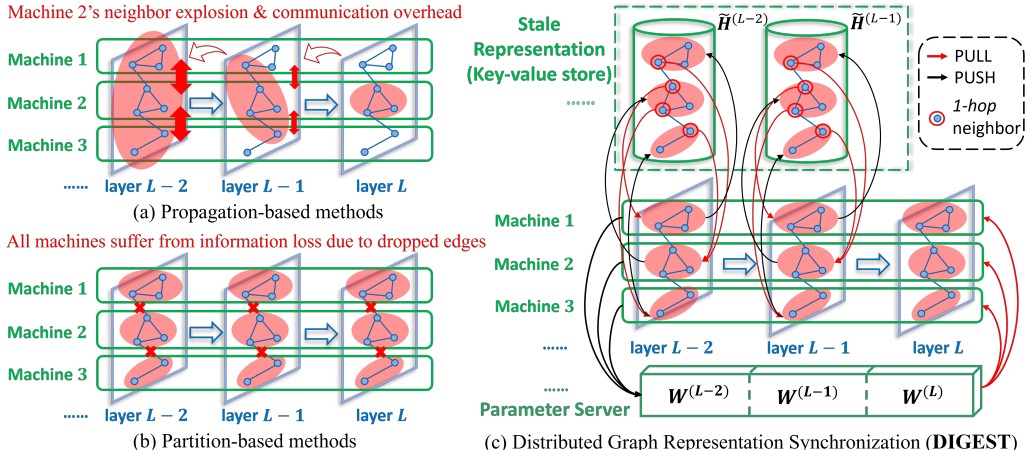

Figure 1: **Distributed GNN training methods. (a)**: Propagation-based methods rely on communication of out-of-subgraph neighbor nodes for exact message passing even in a distributed setup. **(b)**: Partition-based methods decompose the original problem into multiple smaller ones and directly apply data parallelism onto partitioned subgraph data. **(c)**: In DIGEST each device utilizes the stale representations of all its neighbors from other subgraphs. Propagation-based methods suffers high communication cost (red vertical double arrows in (a)) due to neighbor explosion, while partition-based methods suffer severe information loss due to dropped edges (red crosses in (b)). DIGEST combines the best of both worlds. ALL nodes are utilized in DIGEST to achieve full-graph awareness, while periodic stale representation synchronization keeps the communication cost low.

where $\mathbf{W}_m = \{\mathbf{W}_m^{(\ell)}\}_{\ell=1}^{L}$ are local parameters and $\mathbf{h}_v^{(L)}$ follows Eq. 2 recursively.

**Challenges.** The main challenges for distributed training of GNNs lie in the trade-off between *communication cost* and *information loss*. "Partition-based" method generalizes the existing data parallelism techniques of classical distributed training on *i.i.d* data to graph data and enjoys minimal communication cost. However, directly partitioning a large graph into multiple subgraphs can result in severe information loss due to the ignorance of huge number of cross-subgraph edges and cause performance degeneration (Angerd et al., 2020; Jia et al., 2020; Ramezani et al., 2021). For these methods, the representation of neighbors out of the current subgraph (second representation set in Eq. 2) are dropped and the connections between subgraphs are thus ignored. Hence, another line of work (Wang et al., 2019), namely "propagation-based" method considers using communication of neighbor nodes for each subgraph to satisfy GNN's neighbor aggregation, which minimizes the information loss. As shown in Eq. 2, the representations for neighbor nodes outside the current subgraph is swapped between different subgraphs. However, the number of neighbors involved in the neighbor aggregation process expands exponentially as the GNN model goes deep, which is known as the *neighborhood explosion* problem. Hence, though no edges are dropped in this case, inevitable communication overhead is incurred and plagues the achievable training efficiency (Ma et al., 2019; Zhu et al., 2019; Zheng et al., 2020; Tripathy et al., 2020; Wan et al., 2022). Moreover, theoretical guarantees (e.g., on convergence, approximation error) are not well explored for distributed GNN due to the joint sophistication of graph structure and neural network optimization.

## 3 PROPOSED METHOD

In this section, we introduce the proposed GNN training framework DIGEST. DIGEST leverages *both* types of representations in Eq. 2 to address the information loss issue. In addition, instead of exchanging real-time representations during the training process between the subgraphs, DIGEST only pull and push the stale representations before or after each step of training periodically. With this strategy, the communications turn to be more efficient, which are illustrated in Figure 1 and analyzed in more details in Section 3.3. Moreover, we prove that the error introduced by the staleness of the stale representation is upper-bounded while the convergence is also guaranteed.

### 3.1 DISTRIBUTED GNN TRAINING WITH FULL-GRAPH AWARENESS

In DIGEST, each copy of GNN trained on a local machine will make use of all available graph information, i.e. *no* edges are dropped in both forward and backward propagation. Analytically, calculating each local gradient $\nabla \mathcal{L}_m^{\text{Local}}$ as defined in Eq. 3 will involve out-of-subgraph neighbor information. For out-of-subgraph neighbor nodes, we approximate their representations via stale representations acquired in previous training, denoted by $\tilde{\mathbf{h}}_v^{(\ell)}$. Formally, given a node $v \in \mathcal{G}_m(\mathcal{V}_m, \mathcal{E}_m)$, the forward propagation for the $(\ell+1)$-th layer of DIGEST is achieved by modifying Eq. 2 as

$$\mathbf{h}_v^{(\ell+1)} = \Psi^{(\ell+1)}\bigg(\mathbf{h}_v^{(\ell)}, \Phi^{(\ell+1)}\Big(\big\{\mathbf{h}_u^{(\ell)} : u \in \mathcal{N}(v) \cap \mathcal{V}_m\big\} \cup \underbrace{\big\{\tilde{\mathbf{h}}_u^{(\ell)} : u \in \mathcal{N}(v) \setminus \mathcal{V}_m\big\}}_{\text{Stale representation}}\Big)\bigg). \quad (4)$$

As can be seen, DIGEST considers ALL neighbor nodes information during forward propagation. On the other hand, leveraging the entire graph data in forward propagation will in turn improve the estimation of gradient in backpropagation. To see this, we reformulate Eq. 4 into the matrix form:

$$\mathbf{H}_{in}^{(\ell+1,m)} = F\Big(\mathbf{H}_{in}^{(\ell,m)}, \tilde{\mathbf{H}}_{out}^{(\ell,m)}\Big) := \sigma\Big(\mathbf{P}_{in}^{(m)}\mathbf{H}_{in}^{(\ell,m)}\mathbf{W}_m^{(\ell+1)} + \mathbf{P}_{out}^{(m)}\tilde{\mathbf{H}}_{out}^{(\ell,m)}\mathbf{W}_m^{(\ell+1)}\Big), \quad (5)$$

where $\mathbf{H}_{in}^{(\ell,m)}$ and $\tilde{\mathbf{H}}_{out}^{(\ell,m)}$ denotes the matrix of in-subgraph node representations and out-of-subgraph stale representations at $\ell$-th layer on subgraph $\mathcal{G}_m$, respectively. $F$ denotes the forward propagation function of one layer of GNN for compact formula. We consider the GCN model as an example for illustration but our analyses apply to general cases of any GNN models. $\mathbf{P}_{in}^{(m)}$ and $\mathbf{P}_{out}^{(m)}$ denotes the propagation matrix for in-subgraph nodes and out-of-subgraph nodes of $\mathcal{G}_m$, respectively, and we have $\mathbf{P}_m = \mathbf{P}_{in}^{(m)} + \mathbf{P}_{out}^{(m)}$ where $\mathbf{P}_m$ is the original propagation matrix for subgraph $\mathcal{G}_m$. $\sigma(\cdot)$ is the activation function following GCN's definition. Hence, the gradient over model parameters is

$$\frac{\partial}{\partial \mathbf{W}_m^{(\ell+1)}} F\Big(\mathbf{H}_{in}^{(\ell,m)}, \tilde{\mathbf{H}}_{out}^{(\ell,m)}\Big) = \frac{\partial}{\partial \mathbf{W}_m^{(\ell+1)}} \sigma\Big(\mathbf{P}_{in}^{(m)}\mathbf{H}_{in}^{(\ell,m)}\mathbf{W}_m^{(\ell+1)} + \mathbf{P}_{out}^{(m)}\tilde{\mathbf{H}}_{out}^{(\ell,m)}\mathbf{W}_m^{(\ell+1)}\Big)$$
$$= \Big[\mathbf{P}_{in}^{(m)}\mathbf{H}_{in}^{(\ell,m)} + \mathbf{P}_{out}^{(m)}\tilde{\mathbf{H}}_{out}^{(\ell,m)}\Big]^\top \sigma'\Big(\mathbf{P}_{in}^{(m)}\mathbf{H}_{in}^{(\ell,m)}\mathbf{W}_m^{(\ell+1)} + \mathbf{P}_{out}^{(m)}\tilde{\mathbf{H}}_{out}^{(\ell,m)}\mathbf{W}_m^{(\ell+1)}\Big). \quad (6)$$

The key observation here is that ALL neighbor nodes are involved in the backpropagation since the gradient above depends on $\tilde{\mathbf{H}}_{out}^{(\ell,m)}$. The separation of in-subgraph nodes and out-of-subgraph nodes, and their approximation via stale representation form the very foundation of DIGEST.

### 3.2 SYSTEM DESIGN

This section presents the overall system design of DIGEST as depicted in Figure 1. DIGEST maintains a shared-memory-based KVS for storing and retrieving representations. KVS can be easily extended to a truly distributed storage to support large-scale distributed training spanning multiple servers.

We first introduce two operations used by DIGEST to store and retrieve representations. The stale representations of layer $\ell$ for all nodes in $\mathcal{V}$ can be formulated as $\tilde{\mathbf{H}}^{(\ell)} = \{\tilde{\mathbf{h}}_v^{(\ell)} : v \in \mathcal{V}\}$. For any subgraph $\mathcal{G}_m$ to start the forward process of layer $\ell$, the necessary stale representations $\tilde{\mathbf{H}}_{out}^{(\ell,m)} = \{\tilde{\mathbf{h}}_u^{(\ell)} : u \in \mathcal{N}(v) \setminus \mathcal{V}_m, \forall\, v \in \mathcal{V}_m\}$ are pulled from the KVS that stores representations; this is called a "pull" operation denoted as $\mathbf{H}_{out}^{(\ell,m)} \leftarrow \tilde{\mathbf{H}}_{out}^{(\ell,m)}$. See Figure 1(c). After the end of a epoch, the newly-computed representations $\mathbf{H}_{in}^{(\ell,m)} = \{\mathbf{h}_v^{(\ell)} : \forall\, v \in \mathcal{V}_m\}$ are pushed to the KVS, and these newly-stored representations will be fetched as stale representations in future epochs; this is called a "push" operation denoted as $\mathbf{H}_{in}^{(\ell,m)} \rightarrow \tilde{\mathbf{H}}_{in}^{(\ell,m)}$.

DIGEST features two training modes: (1) DIGEST: a synchronous mode designed ideal for homogeneous training environments. (2) DIGEST-A: an asynchronous mode that better fits for a heterogeneous training environment. DIGEST and DIGEST-A follow different parameter and representation updating strategies. In DIGEST, for each global round, before fetching the aggregated parameters and pulling the stale representations, each subgraph has to wait for other subgraphs to finish updating the latest parameters to the parameter server (PS) and their local representations to the KVS. However, some subgraphs may have lower computing resource compared to other

subgraphs, which we call stragglers. This may lead to imbalanced local training times. In this case, with the synchronous mode, the overall training process can be bottlenecked by the slowest subgraph, therefore suffering from prolonged training time. To address this issue, DIGEST-A applies an asynchronous, non-blocking strategy, where each subgraph directly pulls/pushes stale representations of other subgraphs from the shared KVS and downloads/uploads parameters from the PS without blindly waiting for the slowest subgraph to finish. For better scalability, we will explore disaggregated storage techniques (Klimovic et al., 2016; Nanavati et al., 2017; Amaro et al., 2020) as part of our future work, where DIGEST can utilize a network-attached, high-performance far memory storage system for representation storage and retrieval. We summarize our algorithm in Algorithm 1 in the appendix due to limited space.

In addition, DIGEST and DIGEST-A use several optimizations to minimize the I/O overhead introduced by pulls and pushes. First, we observe that there are a large number of node representations involved in both pull and push operations, and more importantly, nodes are independent of each other on these two operations. Hence, it is inherently suitable for parallel I/O at the granularity of node level. For subgraph $\mathcal{G}_m$, the total number of stale representations needed to be pulled from the KVS is $|\tilde{\mathbf{H}}_{out}^{(\ell,m)}|$. Assume that it takes time $t$ to pull the stale representation of one node, the total time cost should be $|\tilde{\mathbf{H}}_{out}^{(\ell,m)}| \times t$ if being pulled in serial. But with parallel I/O where needed representations are pulled in parallel, theoretically we can still keep the pull time for $v \in \mathcal{V}_m$ as $t$. Additionally, we observe that the pull operation for $\tilde{\mathbf{H}}_{out}^{(\ell,m)}$ can be overlapped with the forward process of layer $\ell - 1$; similarly, the push operation for $\mathbf{H}_{in}^{(\ell,m)}$ can be overlapped with the forward process of layer $\ell + 1$. The training process on each subgraph is depicted in Figure 2. The cost of pull/push operations is hidden by the layer forward process, therefore, is eliminated.

Second, to further reduce the I/O overhead, DIGEST uses a periodic representation synchronization strategy, which pushes updated representations to the KVS once every $N$ epochs. This introduces a trade-off in I/O overhead and training performance. Increasing the frequency of the periodic synchronization will benefit performance, but this will introduce more I/O overhead. We analyze this trade-off in Section 5.2.

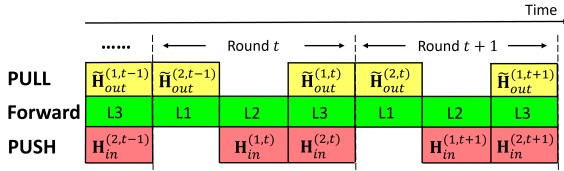

Figure 2: Illustration of DIGEST's concurrent pull/push and forward propagation operations on a 3-layer GNN.

### 3.3 Complexity analyses

Here we analyze the memory and communication complexity of DIGEST. DIGEST pulls the required out-of-subgraph node representations and keeps them locally for each local machine. For the $m$-th local machine and the corresponding subgraph on it, i.e. $\mathcal{G}_m(\mathcal{V}_m, \mathcal{E}_m)$, the memory complexity per training iteration is $\mathcal{O}\big(\big|\bigcup_{v \in \mathcal{V}_m} \mathcal{N}(v) \cup \{v\}\big| Ld\big)$, which scales *linearly* with respect to the number of GNN layers. DIGEST's communication cost per round can be expressed as $\mathcal{O}\big(MLd^2 + \sum_{m=1}^{M} \big|\bigcup_{v \in \mathcal{V}_m} \mathcal{N}(v) \setminus \mathcal{V}_m\big| Ld + NLd\big)$. Again, the communication cost of DIGEST is only *linear* with respect to GNN depth $L$.

## 4 Theoretical analyses

In this section, we provide theoretical analyses of the propose distributed strategy DIGEST, including the bound of error induced by the staleness of node representations, and convergence guarantee for DIGEST under both synchronous and asynchronous settings. All proofs can be found in the appendix.

### 4.1 Error bound on global approximated gradients

Our first theorem shows that under the distributed setting, the approximation error of the global model's gradients can be upper bounded by the staleness of node representations.

**Theorem 1.** *Given a $L$-layer GNN $f_{\mathbf{W}}$ with $r_1$-Lipschitz smooth $\Phi$ and $r_2$-Lipschitz smooth $\Psi$. Denote $\Delta(\mathcal{G})$ as the maximal node degree for graph $\mathcal{G}$. Assume $\forall v \in \mathcal{V}$ and $\forall \ell \in \{1, 2, \cdots, L-1\}$*

we have $\|\mathbf{h}_v^{(\ell)} - \tilde{\mathbf{h}}_v^{(\ell)}\| \leq \epsilon^{(\ell)}$, where $\mathbf{h}_v^{(\ell)}$ and $\tilde{\mathbf{h}}_v^{(\ell)}$ denotes the node representation computed by DIGEST and the stale one, respectively. Further assume each local loss function $\mathcal{L}_m^{Local}$ is $\tau$-Lipschitz smooth w.r.t the node representation. Then, we have that $\left\|\nabla_\mathbf{W}\mathcal{L} - \nabla_\mathbf{W}\mathcal{L}^*\right\|_2 \leq (\tau/M)\sum_{\ell=1}^{L-1}\epsilon^{(\ell)}r_1^{L-\ell}r_2^{L-\ell}\sum_{m=1}^M |\Delta(\mathcal{G}_m)|^{L-\ell}$, where $\nabla_\mathbf{W}\mathcal{L}$ and $\nabla_\mathbf{W}\mathcal{L}^*$ denotes the global gradient computed by DIGEST and the exact global gradient without any staleness.

## 4.2 CONVERGENCE OF SYNCHRONOUS DIGEST

As both fresh inner-subgraph and stale out-of-subgraph representations are adopted in our algorithm, its convergence rate is still unknown. We have proved the convergence of DIGEST and present the convergence property in the theorem blow. First, we introduce some assumptions:

**Assumption 1.** *The loss function $Loss(\cdot, \cdot)$ is $C_{Loss}$-Lipchitz continuous and $L_{Loss}$-Lipschitz smooth with respect to the last layer's node representation, i.e., $|Loss(\mathbf{h}_v^{(L)}, \mathbf{y}_v) - Loss(\mathbf{h}_w^{(L)}, \mathbf{y}_v)| \leq C_{Loss}\|\mathbf{h}_v^{(L)} - \mathbf{h}_w^{(L)}\|_2$ and $\|\nabla Loss(\mathbf{h}_v^{(L)}, \mathbf{y}_v) - \nabla Loss(\mathbf{h}_w^{(L)}, \mathbf{y}_v)\|_2 \leq L_{Loss}\|\mathbf{h}_v^{(L)} - \mathbf{h}_w^{(L)}\|_2$.*

**Assumption 2.** *The activation function $\sigma(\cdot)$ is $C_\sigma$-Lipchitz continuous and $L_\sigma$-Lipschitz smooth, i.e. $\|\sigma(Z_1^{(\ell)}) - \sigma(Z_2^{(\ell)})\|_2 \leq C_\sigma\|(Z_1^{(\ell)} - Z_2^{(\ell)})\|_2$ and $\|\sigma'(Z_1^{(\ell)}) - \sigma'(Z_2^{(\ell)})\|_2 \leq L_\sigma\|(Z_1^{(\ell)} - Z_2^{(\ell)})\|_2$.*

**Assumption 3.** *$\forall\, \ell = 1, 2, \cdots, L$, we have $\|W^{(\ell)}\|_F \leq K_W$, $\|P^{(\ell)}\|_P \leq K_W$, $\|X^{(\ell)}\|_F \leq K_X$.*

**Theorem 2.** *Consider GCN with $L$ layers that is $L_f$-Lipschitz smooth. $\forall\, \epsilon > 0$, $\exists$ constant $E > 0$ such that, we can choose a learning rate $\eta = \frac{\sqrt{M}\epsilon}{E}$ and number of training iterations $T = (\mathcal{L}(\mathbf{W}^{(1)}) - \mathcal{L}(\mathbf{W}^*))\frac{E}{\sqrt{M}}\epsilon^{-\frac{3}{2}}$ s.t., $T^{-1}\sum_{t=1}^T \|\nabla\mathcal{L}(\mathbf{W}^{(t)})\|^2 \leq \mathcal{O}(T^{-2/3}M^{-1/3})$, where $\mathbf{W}^*$ denotes the optimal parameter.*

Our convergence rate of DIGEST is $\mathcal{O}(T^{-2/3}M^{-1/3})$, which is better than pipeline-parallelism method $\mathcal{O}(T^{-2/3})$ (Wan et al., 2022) and sampling-based method $\mathcal{O}(T^{-1/2})$ (Chen et al., 2018; Cong et al., 2021), and very close to full-graph training $\mathcal{O}(T^{-1})$.

## 4.3 CONVERGENCE OF ASYNCHRONOUS DIGEST

Convergence for asynchronous distributed algorithms could be even harder to obtain due to the delay in parameter's update (the global model's parameters may have been updated several times when the slowest local machine finishes its computation.) Our main result is shown below:

**Assumption 4.** *$\forall\, m \in [M]$, $\|\nabla\tilde{\mathcal{L}}_m(W)\|_2 \leq V \cdot \|\nabla\mathcal{L}(W)\|_2$, and $\langle\nabla\mathcal{L}(W), \nabla\tilde{\mathcal{L}}_m(W)\rangle \geq \beta \cdot \|\nabla\mathcal{L}(W)\|_2^2$, where $V$ and $\beta$ are positive real numbers, i.e., $V, \beta \in \mathbb{R}^+$.*

**Theorem 3.** *Assume the global model $\mathcal{L}(W)$ is $C_f$-Lipschitz continuous and the delay is bounded, i.e., $\tau < K$. Further, assume $\beta - \frac{V^2}{2} > 0$. There exist constant $B$ and a second-order polynomial of learning rate $\eta$, i.e., $P(\eta)$ such that after $T$ global iterations on the server, asynchronous DIGEST converges to the optimal parameter $\mathbf{W}^*$ by $T^{-1}\sum_{t=1}^T \left\|\nabla\mathcal{L}(\mathbf{W}^{(t)})\right\|_2^2 \leq (\eta TB)^{-1}(\mathcal{L}(\mathbf{W}^{(1)}) - \mathcal{L}(\mathbf{W}^{(*)})) + P(\eta)/B$, where $B = \beta - \frac{V^2}{2}$ and $P(\eta) = \frac{1}{2}\eta^2 K^2 C_f^2 L_f^2 + (1+V)\eta K C_f^2 L_f$.*

## 5 EXPERIMENTS

In this section, we evaluate DIGEST and compare DIGEST against two state-of-the-art distributed GNNs training frameworks as baselines in terms of training efficiency and scalability. Considering the distinct training time per epoch between DIGEST and other baselines, we report the F1 scores on validation dataset and training loss over training time, instead of over communication rounds, in the results. This way it makes a fairer comparison in terms of training performance and efficiency.

### 5.1 EXPERIMENT SETTING

**Implementation and Setup.** We have implemented DIGEST and other comparison GNNs training methods all in PyTorch (Paszke et al., 2019). For all the experiments, we simulate a distributed training environment using an EC2 `g4dn.metal` virtual machine (VM) instance on AWS, which has 8 NVIDIA T4 GPUs, 96 vCPUs, and 384 GB main memory. We implemented the shared-memory KVS using the Plasma in-memory object store[3] for representation storage and retrieval.

---

[3] https://arrow.apache.org/docs/python/plasma.html

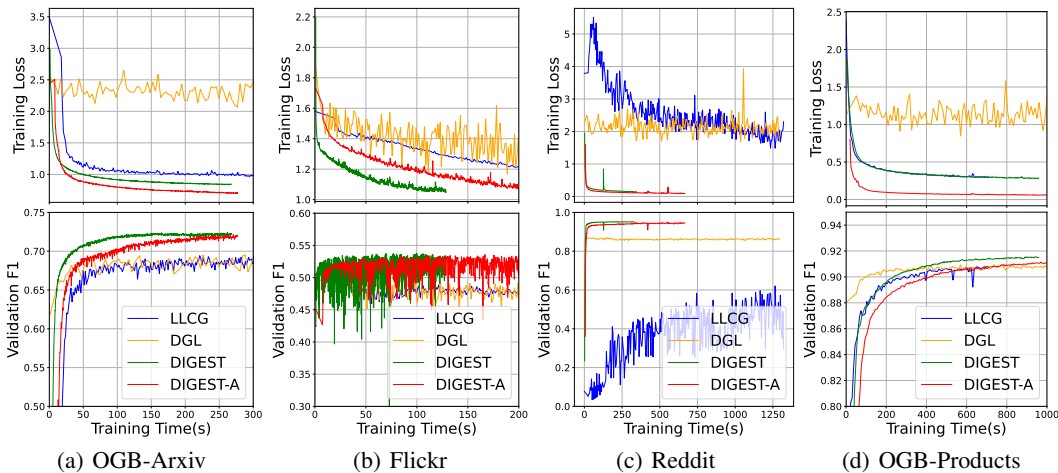

| (a) OGB-Arxiv | (b) Flickr | (c) Reddit | (d) OGB-Products |

Figure 3: Performance comparison of the GCN training frameworks on four benchmark datasets. The top four subfigures show the training loss over training time, and the bottom four subfigures show the global validation F1 scores during the whole training process. (Best viewed in color.)

Table 1: Performance comparison of distributed GNNs frameworks. F1 score on validation dataset reported. Speedup is calculated by normalizing per-epoch training time against that of DGL.

| Method | Metric | GCN | | | | GAT | | |
|---|---|---|---|---|---|---|---|---|
| | | OGB-Arxiv | Flickr | Reddit | OGB-Products | OGB-Arxiv | Flickr | Reddit |
| LLCG | F1 | $69.8 \pm 0.21$ | $50.73 \pm 0.15$ | $62.09 \pm 0.41$ | $90.79 \pm 0.16$ | $68.84 \pm 0.22$ | $43.98 \pm 0.32$ | $91.1 \pm 0.17$ |
| | Speedup | $2.35\times$ | $0.88\times$ | $1.47\times$ | $1.396\times$ | $1.787\times$ | $0.923\times$ | $9.956\times$ |
| DGL | F1 | $69.9 \pm 0.17$ | $50.9 \pm 0.13$ | $87.02 \pm 0.23$ | $91.01 \pm 0.12$ | $70.34 \pm 0.17$ | $51.50 \pm 0.27$ | $92.58 \pm 0.12$ |
| | Speedup | $1\times$ | $1\times$ | $1\times$ | $1\times$ | $1\times$ | $1\times$ | $1\times$ |
| **DIGEST** | F1 | $72 \pm 0.23$ | $53.78 \pm 0.21$ | $95.23 \pm 0.43$ | $91.55 \pm 0.1$ | $68.35 \pm 0.41$ | $52.08 \pm 0.21$ | $94.19 \pm 0.15$ |
| | Speedup | $17.41\times$ | $11.06\times$ | $7.86\times$ | $3.096\times$ | $11.49\times$ | $6.591\times$ | $21.817\times$ |
| **DIGEST-A** | F1 | $71.9 \pm 0.16$ | $53.1 \pm 0.32$ | $94.55 \pm 0.37$ | $91.54 \pm 0.1$ | $69.04 \pm 0.13$ | $52.16 \pm 0.17$ | $93.95 \pm 0.22$ |

**Baselines.** Recall in Section 2 we categorize existing distributed GNN training into two types of general methods. In evaluation, we choose two state-of-the-art distributed training frameworks, one from each category as the baseline. For the first category, we choose LLCG (Ramezani et al., 2021), which partitions a graph into subgraphs and trains each subgraph strictly independently without incurring any communication among subgraphs. LLCG uses a central server to aggregate local models from each device and performs global training using mini-batches with full neighbor information to ensure that the model learns the global structure of the graph. LLCG uses this additional step to reduce the information loss caused by graph partitioning. For the second category, we choose to use DGL (Wang et al., 2019), which is a commonly-used, distributed GNN training framework. In contrast to LLCG, DGL requires exchanging node representations among partitioned subgraphs. DGL requires frequent swap operations with other subgraphs for representations during subgraph's local training in each epoch, and therefore, DGL incurs high communication cost.

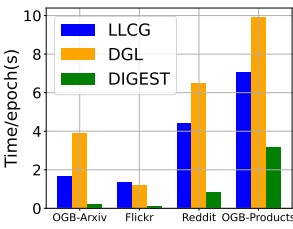

Figure 4: Training time/epoch.

Figure 5: Scalability.

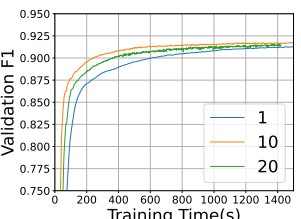

Figure 6: Synchronization interval. Better seen in color.

## 5.2 EXPERIMENTAL RESULTS

In this section, we evaluate both sync & async versions of DIGEST, LLCG, and DGL on the four datasets. Due to the page limit, we move parts of our evaluation results to the Appendix.

**Efficiency of DIGEST.** We first evaluate the training performance of DIGEST and DIGEST-A. As shown in Figure 3, DIGEST outperforms both LLCG and DGL for all the datasets when performing distributed training on a pure GCN. LLCG performs worst particularly for the Reddit dataset, because in the global server correction of LLCG, only a mini-batch is trained and it is not sufficient to correct the plain GCN. This is also the reason why the authors of LLCG report the performance of a complex model with mixing GCN layers and GraphSAGE layers Ramezani et al. (2021). DGL achieves good performance on some dataset (e.g., OGB-products) with uniform node sampling strategy and real-time representation exchanging. However, frequent communication also leads to slow performance increasing for dataset Flickr (Figure 3(b)) and poor performance for all four datasets. DIGEST and DIGEST-A avoid these issues and therefore achieve satisfying performance over the training time. DIGEST-A is slowly catching up DIGEST due to the diverse model parameters used by subgraphs in the early training period.

We measure the training time per epoch as shown in Figure 4. Since the representation synchronization is only performed before the start or after the end of local training, DIGEST takes significantly shorter training time per epoch than that of LLCG and DGL. Furthermore, DIGEST performs periodic synchronization instead of per-epoch synchronization, which further shortens the training time.

Table 1 presents the detailed numbers for the comparison of three frameworks on the four datasets. For all the cases except GAT on OGB-Arxiv, DIGEST achieves leading F1 scores on the validation dataset, demonstrating the efficacy of DIGEST's design.

**Scalability of DIGEST.** We evaluate the scalability of three frameworks by training a GCN on OGB-Products with varied number of GPUs. We use average training time per epoch against that of DGL with a single GPU to calculate the speedup results. As shown in Figure 5, DIGEST shows the best scalability compared to the other two. The speedup rises with the number of GPUs used during training. We observe a similar trend for DGL, but the relative speedup for DGL is significantly smaller than that for DIGEST, due to the using of real-time representations instead of stale representations.

**Synchronization frequency.** We next perform a sensitivity analysis by varying the synchronization intervals for OGB-Products to study how the synchronization frequency would affect the training performance. As shown in Figure 6, DIGEST achieves the highest F1 score over training time when configured to perform synchronization of stale representations every 10 epochs. A large interval (20) or a small interval (1) results in performance degradation, due to the long term loss of graph information or additional communication cost.

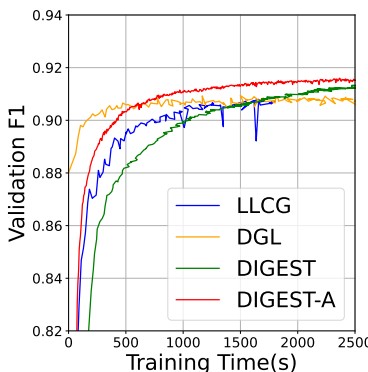

Figure 7: Performance comparison of OGB-Products when trained in a heterogeneous environment.

**Training in heterogeneous environment.** Finally, we test DIGEST's asynchronous training mode. As stated in Section 3.2, asynchronous training is better suited for GNN training in heterogeneous environments. In this test, we randomly select one subgraph as the straggler before the training starts. To simulate the straggler lagging caused by limited computing capability, a random delay ranging from 8 to 10 seconds is added to the chosen straggler during the whole training process. We can see from Figure 7 that DIGEST-A performs much better than other three synchronous methods and converge to high F1-score at the early stage of the training. This is because asynchronous mode effectively eliminates GPU's blocking caused by waiting with significantly improved GPU utilization.

## 6 CONCLUSION

There are two general categories in distributed GNN training. Partition-based methods suffer from graph information loss, while propagation-based methods suffer from high communication cost. In this work we present DIGEST, a novel distributed GNN training framework that synergizes the complementary strengths of both methods by leveraging stale representations intelligently. We provide rigorous theoretical analysis to prove that DIGEST has competitive convergence rate and bounded error due to staleness. Extensive experiments on four benchmark datasets validate our analysis and demonstrate the efficiency and scalability of DIGEST.

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

# A  APPENDIX

In this section, we clarify our contributions made in this work and describe detailed experimental setup, additional experimental results, and complete proofs. We reuse part of code adopted from GNNAutoScale Fey et al. (2021); our code is available at: `https://anonymous.4open.science/r/DIGESTA-78F2/`. Please note that the code is subjected to reorganization to improve the readability.

## A.1  CONTRIBUTIONS AND NOVELTY

The contributions and novelty of this work are multi-fold. In this paper, we propose a new, highly-parallel, and full-graph-aware distributed GNN training method; on top of this new method, we design a novel, compute-and-storage-disaggregated training system to enable better scalability and allow distributed GNN training to potentially benefit from emerging computing paradigms and hardware; finally, we deduce new theoretical guarantees and analyses for the co-designed algorithms and systems.

**(1).  Methodology Novelty in Algorithm-System Co-design:** Our paper is mainly motivated from a distributed training perspective, where the proposed framework synergizes the best of both partition-based and propagation-based distributed training; GNNAutoScale provides a theoretical foundation, which exposes potential opportunities that can be harnessed by and *co-designed with* new distributed training system infrastructures to enable highly-parallel GNN training. *DIGEST goes beyond GNNAutoScale in that we built a novel distributed training framework that effectively decouples the management of state (i.e., representations) and compute (i.e., GNN training).*

**(2). System Architecture Novelty:** The disaggregated architecture of DIGEST is the result of an algorithm-system co-design as mentioned in the Methodology Novelty, and enables great properties including high scalability and low training time, as demonstrated in our paper. More importantly, this disaggregated architecture could enable fundamental opportunities for GNN training systems to take advantage of emerging computing paradigm such as elastic serverless computing as well as emerging hardware such as Zoned Namespace SSD (ZNS) and smart programmable network hardware (SmartNIC); in this work, we have shown the promising scalability and speedup that DIGEST offers, which establishes a solid system foundation for further system-level optimizations and innovations. This demands/inspires future research along the line, which we plan to do as part of our future work.

**(3). Theoretical Novelty:** All of our theoretical analyses are tailored for a distributed training setup, while GNNAutoScale only considers single-GPU training.

## A.2  EXPERIMENTAL SETUP DETAILS

As mentioned in Section 5.1, all the experiments are done on an EC2 `g4dn.metal` virtual machine (VM) instance on AWS, which has 8 NVIDIA T4 GPUs, 96 vCPUs, and 384 GB main memory. Other important information including operation system version, Linux kernel version, and CUDA version is summarized in Table 2. For fair comparison, we use the same optimizer (Adam), learning rate, and graph partition algorithm for all the three frameworks, DIGEST, LLCG, and DGL. For parameters that are unique to both LLCG and DGL, such as the number of neighbors sampled from each layer for each node, we choose the default value for both LLCG and DGL. Each of the three frameworks has a set of parameters that are exclusively unique to that framework; for these exclusive parameters, we tune them in order to achieve the best performance. Please refer to the configuration files under `run/conf/model` for detailed configuration setups for all the models and datasets.

Table 2: Summary of environmental setup of our testbed.

| OS | Linux kernel | CUDA | Driver | PyTorch | PyTorch Geometric | PyTorch Sparse |
|---|---|---|---|---|---|---|
| Ubuntu 18.04 | 5.4.0 | 11.6 | 510.47.03 | 1.10.0 | 2.0.4 | 0.6.13 |

We use four datasets: OGB-Arxiv Hu et al. (2020), Flickr Zeng et al. (2019), Reddit Zeng et al. (2019), and OGB-Products Hu et al. (2020) for evaluation. The detailed information of these datasets is summarized in Table 3.

Table 3: Summary of dataset statistics.

| Dataset | # Nodes | # Edges | # Features | # Classes | Train % / Validation % / Test % |
|---|---|---|---|---|---|
| Flickr | 89,250 | 899,756 | 500 | 7 | 50% / 25% / 25% |
| Reddit | 232,965 | 23,213,838 | 602 | 41 | 66% / 10% / 24% |
| OGB-Arixv | 169,343 | 2,315,598 | 128 | 40 | 53.7% / 17.6% / 28.7% |
| OGB-Products | 2,449,029 | 123,718,280 | 100 | 47 | 8% / 2% / 90% |

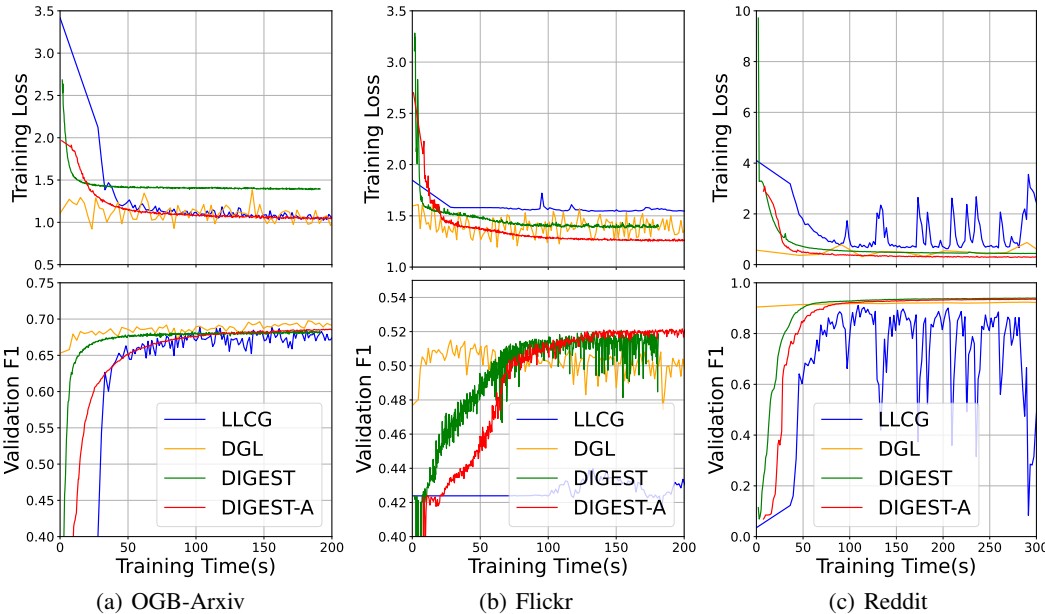

Figure 8: Performance comparison of different distributed GAT training methods on four benchmark datasets. The top three subfigures show the training loss during the whole training process; the bottom three subfigures show the global validation F1 scores during the whole training process.

## A.3 ADDITIONAL EXPERIMENTAL RESULTS

### A.3.1 PERFORMANCE OF GAT TRAINING

We first show the learning curves of training GAT with three methods on three different datasets. As shown in Figure 8, for dataset Flickr and Reddit, DIGEST acheives the best validation F1 score over training time of all the three frameworks. For dataset OGB-Arxiv, the performance of DIGEST is slightly worse than DGL but still outperforms LLCG. Specifically, LLCG's training curves are not stable and fluctuate dramatically for both GCN and GAT on Reddit. This is because Reddit is much denser compared to other datasets, and in this case, the sampling process of the global server correction in LLCG has difficulty capturing all the information loss due to the cut-edges. Unlike LLCG, DIGEST's training curves are much smoother not only for GCN training but also for GAT training.

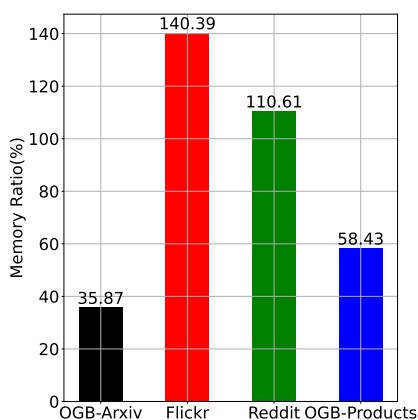

Figure 9: The average ration of the number of out-of-subgraph nodes to the number of in-subgraph nodes when training a GCN over the four datasets.

### A.3.2 MEMORY OVERHEAD

In this section we quantify the memory overhead introduced by DIGEST.

**Ratio of out-of-subgraph nodes and in-subgraph nodes.**
Figure 9 shows the ratios of the number of out-of-subgraph nodes to the number of in-subgraph nodes across four datasets. This ratio quantifies additional memory consumption compared to methods that does not use any information of the neighboring nodes during training.

Denser graphs like Flickr and Reddit require more memory to store representations of off-subgraph nodes than OGB-Arxiv and OGB-products. This introduces an interesting tradeoff between extra memory storage and gained benefits in reduced communication and preservation of global graph information. We argue that modern GPU servers are equipped with ample GPU memory resources to buffer the out-of-subgraph representations Gandhi & Iyer (2021); Wang et al. (2021); if indeed more memory is required, our research will benefit from the recent advancement of unified memory Choi et al. (2022); Huang et al. (2020); Peng et al. (2020), where DIGEST can use both the host and GPU memory more efficiently.

In the worst case, if GPU memory is limited, DIGEST can implement a multi-tier storage system that uses the limited memory as a level-one cache and the host memory as a backing store. For large graphs that are sparse (OGB-products), the extra memory cost can be bounded to a relatively lower ratio (58.43%).

**Host memory cost of stale representations.** The KVS is responsible for storing the representations of all the nodes in a graph. The representations are stored in the memory of the host server instead of the GPUs, the latter of which is rather limited. We implemented the in-memory KVS with Apache Plasma, which is a shared memory storage that supports efficient, shared-memory-based inter-process communication (IPC) for multiple training processes located on the same server. However, extending our current KVS implementation to a fully-distributed storage system is trivial. Using off-the-shelf, high-performance distributed in-memory KVSes such as Redis is one option. Alternatively, we could also implement a simple client library, which can be used by the training process for key-value item mapping (e.g., using the commonly-used consistent hashing algorithm) and remote representation retrieval/storage, and with the client library, we could deploy a cluster of Plasma storage processes either on a dedicated storage cluster or on the same training server cluster to support distributed representation storage.

The overall memory consumption required to store representation data can be calculated with the following equation:

$$KVS\ memory\ usage = (L-1) \times dim \times |V| \times s \quad (7)$$

where $L$ is the total number of layers of the model, $dim$ denotes the hidden dimension, $|V|$ represents the number of nodes in the graph, and $s$ is the size of data type in Python numpy. For the $float32$ data type, it takes 4 bytes for each single value. With the provided formula, for a 3-layer GNN model, training large graph dataset such as OGB-Products (with $2,449,029$ nodes and 128 hidden dimensions), the extra host memory consumption for the representations is around $2*128*2449029*4/1024/1024/1024 = 2.336$ GB. DIGEST exhibits an interesting tradeoff: it uses a small amount of extra memory overhead for storing stale representations to enable the disaggregation of the compute and storage for higher scalability and more flexibility. We also argue that a small host memory cost of several GBs is negligible considering today's multi-GPU servers are equipped with hundreds of GBs if not more than a few TBs of host memory[4].

Table 4: GPU memory consumption.

| Model | OGB-Arxiv | OGB-Products |
|---|---|---|
| GraphSAGE | 0.40 GB | 0.92 GB |
| DGL | 0.64 GB | 1.78 GB |
| LLCG | 0.23 GB | 0.36 GB |
| DIGEST | 0.22 GB | 0.36 GB |

---

[4]For example, AWS EC2's `p3.16xlarge` is equipped with 8 Nvidia Tesla V100 GPUs with 488 GBs of host memory: https://aws.amazon.com/blogs/aws/new-amazon-ec2-instances-with-up-to-8-nvidia-tesla-v100-gpus-p3/.

**GPU memory consumption.** For the concern of GPU memory consumption, we compare DIGEST with GraphSAGE, LLCG, and DGL by including all the information inside a GNN's receptive field in a single optimization step. The comparison results in Table 4 show that DIGEST has the lowest GPU memory consumption across all four systems.

### A.3.3 EMPIRICAL VALIDATION OF GRADIENT APPROXIMATION ERROR

In this section, we empirically evaluate the gradient approximation error due to the usage of stale representation. We conduct this experiment to show that the actual approximation error of gradients of DIGEST compared with the ground-truth gradients (i.e., gradients calculated without any stale representation) can be negligible in practice.

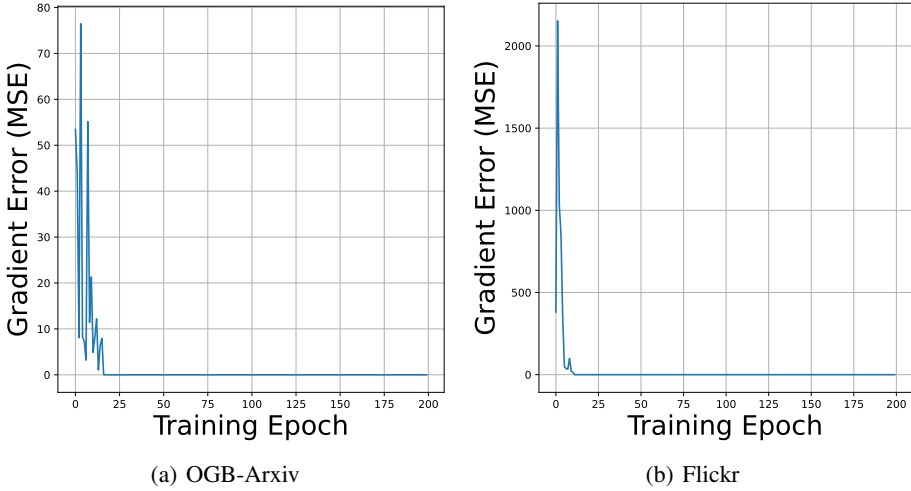

(a) OGB-Arxiv                    (b) Flickr

Figure 10: Error between the gradients calculated by DIGEST and full-graph baseline (i.e., without any staleness). Zoom in for detail.

As can be seen in Figure 10, during the training phase the gradient calculated by DIGEST quickly converge to the ground-truth gradients typically after fewer than 10-20 epochs. Hence, for the majority of training epochs the error of gradients is very small and the impact is negligible, which in turn validate our theoretical analyses in Theorem 1.

### A.3.4 OTHER COMPARISONS

In this section, we compare DIGEST/DIGEST-A with PipeGCN and GNNAutoScale. We train OGB-Products with DIGEST/DIGEST-A, PipeGCN and GNNAutoScale in the heterogeneous environment mentioned in Section 5.2, and report the training time taken to reach the target validation F1 score and time per epoch in Firgure 11, since there is no "epoch" in an asynchronous setting, the value of time per epoch for DIGEST-A is omitted. We can see that DIGEST gets slight higher time per epoch than GNNAutoScale but reduces the time per epoch by $24.13\%$ compared with PipeGCN. Meanwhile, DIGEST-A gets the lowest training time to reach the target F1 score and saves $48.98\%$ and $19.12\%$ training time compared with PipeGCN and GNNAutoScale, respectively.

We further evaluate DIGEST, PipeGCN and GNNAutoScale on a large graph OGB-papers100m which consist of 111 million nodes 1.6 billion edges to show the efficiency of DIGEST. The experiments are done in a homogeneous environment with 32 GPUs. Since mini-batches in GNNAutoScale are trained in a serial manner instead of a parallel distributed setting, only one GPU is used. DIGEST reduces the time per epoch by $21.13\%$ compared with distributed GNN training algorithm PipeGCN.

### A.4 ALGORITHM

Algorithm 1 shows the process of DIGEST's synchronous mode. At the beginning of training, the original graph is partitioned into several subgraphs with off-the-shelf graph clustering methods;

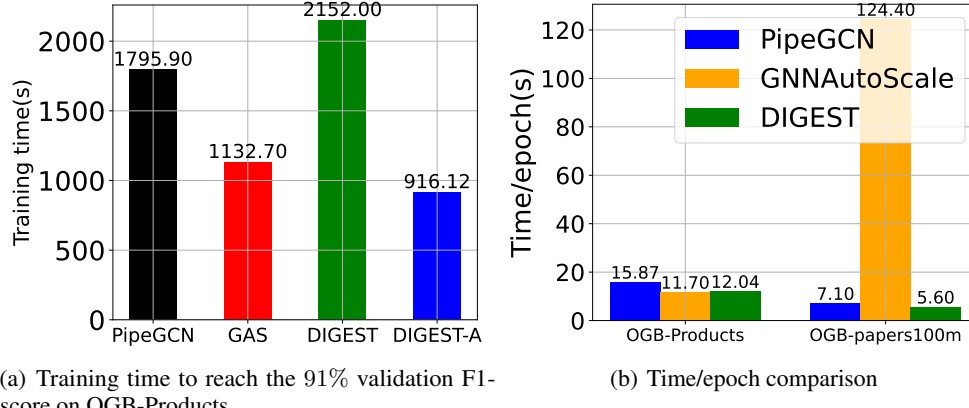

(a) Training time to reach the $91\%$ validation F1-score on OGB-Products

(b) Time/epoch comparison

Figure 11: Time comparisons with PipeGCN and GNNAutoScale.

DIGEST uses the widely-used METIS algorithm Karypis & Kumar (1998). Then the mini-batches are distributed to distinct workers, each of which handles training on a GPU device. Depending on the size of the mini-batch, a single worker can handle one or multiple subgraphs. DIGEST has two types of I/Os: storing and retrieving the model weights and stale representation as illustrated in Figure 1(c). The former one is performed for each epoch by aggregating all the model weights of other subgraphs in parallel (Line 13). For the stale representation synchronization, we synchronize every $N$ epochs, where we empirically tune $N$ to obtain the optimal performance over training time with the defined pull and push operations (Lines 5,6,9,10). To support the asynchronous mode (DIGEST-A), we can simply remove the loop of training epoch and move the parameter aggregation (Line 13) into the subgraph loop.

## A.5 THEORETICAL PROOF

In this section, we provide the formal proof for all the theories presented in the main paper.

### A.5.1 PROOF OF THEOREM 1

**Theorem 4** (Formal version of Theorem 1). *Given a L-layer GNN $f_{\mathbf{W}}$ with $r_1$-Lipschitz smooth $\Phi$ and $r_2$-Lipschitz smooth $\Psi$. Denote $\Delta(\mathcal{G})$ as the maximal node degree for graph $\mathcal{G}$. Assume $\forall\, v \in \mathcal{V}$ and $\forall\, \ell \in \{1, 2, \cdots, L-1\}$ we have $\|\mathbf{h}_v^{(\ell)} - \tilde{\mathbf{h}}_v^{(\ell)}\| \leq \epsilon^{(\ell)}$, where $\mathbf{h}_v^{(\ell)}$ and $\tilde{\mathbf{h}}_v^{(\ell)}$ denotes the node representation computed by DIGEST and the stale one, respectively. Further assume each local loss function $\mathcal{L}_m^{Local}$ is $\tau$-Lipschitz smooth w.r.t node representation. Then the global gradient computed by DIGEST has the following error bound*

$$\left\|\nabla_{\mathbf{W}}\mathcal{L} - \nabla_{\mathbf{W}}\mathcal{L}^*\right\|_2 \leq \frac{\tau}{M}\sum_{\ell=1}^{L-1}\epsilon^{(\ell)}r_1^{L-\ell}r_2^{L-\ell}\sum_{m=1}^{M}|\Delta(\mathcal{G}_m)|^{L-\ell}, \tag{8}$$

*where*

$$\nabla_{\mathbf{W}}\mathcal{L} := \frac{1}{M}\sum_{m=1}^{M}\frac{1}{|\mathcal{V}_m|}\sum_{v\in\mathcal{V}_m}\nabla_{\mathbf{W}}\mathcal{L}_m^{Local}(\mathbf{h}_v^{(L)}), \tag{9}$$

*and*

$$\nabla_{\mathbf{W}}\mathcal{L}^* := \frac{1}{M}\sum_{m=1}^{M}\frac{1}{|\mathcal{V}_m|}\sum_{v\in\mathcal{V}_m}\nabla_{\mathbf{W}}\mathcal{L}_m^{Local}(\mathbf{h}_v^{*(L)}), \tag{10}$$

*where $\mathbf{h}_v^{*(\ell)}$ denotes the exact output from the $\ell$-th layer of GNN without any staleness.*

*Proof.* As stated in Theorem 2 in Fey et al. (2021), under the single-GPU training setup, with Lipschitz smooth $\Phi$ and $\Psi$ as well as not too stale node representations, the GNN last layer's output

---

**Algorithm 1** Distributed GNN training with periodic stale representation synchronization

---

**Input:** Graph $\mathcal{G}(\mathcal{V}, \mathcal{E})$; GNN depth $L$; training epoch $R$; global parameters $\mathbf{W}^{(r)} = \{\mathbf{W}^{(r,\ell)}\}_{\ell=1}^{L}$,
    local parameters $\mathbf{W}_m^{(r)} = \{\mathbf{W}_m^{(r,\ell)}\}_{\ell=1}^{L}, \forall\, m \in [M], r \in [R]$; non-linearity activation
    function $\sigma$; neighborhood function $\mathcal{N} : v \to 2^{\mathcal{V}}$; synchronization interval N; learning rate $\eta$.
**Output:** The trained model weights $\mathbf{W}^{(R+1)}$.

1   `DIGEST():`
2     Initialize $\mathbf{W}^{(1)}$
      $\{\mathcal{G}_m(\mathcal{V}_m, \mathcal{E}_m), m = 1, 2, .., M\} \leftarrow \text{METIS}(\mathcal{G})$                ▷ graph partition
      **for** $r = 1...R$ **do**
3         **for** $m = 1, \cdots, M$ *in parallel* **do**
4            $\mathbf{W}_m^{(r)} = \mathbf{W}^{(r)}$
            **for** $\ell = 1...L$ **do**
5               **if** $r \,\% \, N == 0$ *and* $\ell \neq L$ **then**
6                 $\mathbf{H}_{out}^{(\ell,m)} \leftarrow \tilde{\mathbf{H}}_{out}^{(\ell,m)}$                  ▷ PULL

7               **for** $v \in \mathcal{V}_m$ **do**
8                 $\mathbf{h}_{out}^{(\ell)} = \{\mathbf{h}_u^{(\ell)} : u \in \mathcal{N}(v) \setminus \mathcal{V}_m\}$
                 $\mathbf{h}_{in}^{(\ell)} = \{\mathbf{h}_u^{(\ell)} : u \in \mathcal{N}(v) \cap \mathcal{V}_m\}$
                 $\mathbf{h}_v^{(\ell)} = \sigma\Big(\mathbf{W}_m^{(r,\ell)} \cdot \text{CONCAT}\big(\mathbf{h}_v^{(\ell)}, \mathbf{h}_{in}^{(\ell)}, \mathbf{h}_{out}^{(\ell)}\big)\Big)$

9               **if** $(r-1) \,\% \, N == 0$ *and* $\ell \neq L$ **then**
10                $\mathbf{H}_{in}^{(\ell,m)} \to \tilde{\mathbf{H}}_{in}^{(\ell,m)}$                 ▷ PUSH
11              $\mathbf{h}_v^{(\ell)} \leftarrow \mathbf{h}_v^{(\ell)}/\|\mathbf{h}_v^{(\ell)}\|_2, \forall\, v \in \mathcal{V}_m$      ▷ representation normalization
12              $\mathbf{W}_m^{(r,\ell+1)} = \mathbf{W}_m^{(r,\ell)} - \eta \cdot \bigtriangledown \mathbf{W}_m^{(r,\ell)}$       ▷ update local parameters
13         $\mathbf{W}^{(r+1)} \leftarrow \mathbf{AGG}(\mathbf{W}_1^{(r+1)}...\mathbf{W}_M^{(r+1)})$       ▷ update global parameters
14       **return** $\mathbf{W}^{(R+1)}$

---

can be bounded by

$$\left\| \mathbf{h}_v^{(L)} - \mathbf{h}_v^{*(L)} \right\|_2 \leq \sum_{\ell=1}^{L-1} \epsilon^{(\ell)} r_1^{L-\ell} r_2^{L-\ell} |\mathcal{N}(v)|^{L-\ell}. \tag{11}$$

Now consider the distributed GNN training setting. First, notice that in our distributed setting, the stale node representation $\tilde{\mathbf{h}}_v^{(\ell)}$ is shared for all subgraphs. In other words, for $m = 1, 2, \cdots, M$ we can apply the conclusion above with the Lipschitz smooth asumption and have

$$\begin{aligned}
\left\| \nabla_{\mathbf{W}} \mathcal{L}_m^{\text{Local}}(\mathbf{h}_v^{(L)}) - \nabla_{\mathbf{W}} \mathcal{L}_m^{\text{Local}}(\mathbf{h}_v^{*(L)}) \right\|_2 &\leq \tau \left\| \mathbf{h}_v^{(L)} - \mathbf{h}_v^{*(L)} \right\|_2 \\
&\leq \tau \cdot \sum_{\ell=1}^{L-1} \epsilon^{(\ell)} r_1^{L-\ell} r_2^{L-\ell} |\mathcal{N}(v)|^{L-\ell}.
\end{aligned} \tag{12}$$

Notice that $|\mathcal{N}(v)| \leq \Delta(\mathcal{G}_m), \forall\, v \in \mathcal{V}_m$, where $\Delta(\mathcal{G}_m)$ is defined as the maximal node degree for subgraph $\mathcal{G}_m$. We can sum over all nodes $v \in \mathcal{V}_m$ and take average on both sides of Eq. 12 to get

$$\frac{1}{|\mathcal{V}_m|} \sum_{v \in \mathcal{V}_m} \left\| \nabla_{\mathbf{W}} \mathcal{L}_m^{\text{Local}}(\mathbf{h}_v^{(L)}) - \nabla_{\mathbf{W}} \mathcal{L}_m^{\text{Local}}(\mathbf{h}_v^{*(L)}) \right\|_2 \leq \tau \cdot \sum_{\ell=1}^{L-1} \epsilon^{(\ell)} r_1^{L-\ell} r_2^{L-\ell} |\Delta \mathcal{G}_m|^{L-\ell}. \tag{13}$$

Finally, since we apply average to aggregate each local subgraph's gradient to get the global gradients, by the triangle inequality, we have

$$
\begin{aligned}
&\left\| \nabla_{\mathbf{W}} \mathcal{L} - \nabla_{\mathbf{W}} \mathcal{L}^* \right\|_2 \\
&= \left\| \frac{1}{M} \sum_{m=1}^{M} \frac{1}{|\mathcal{V}_m|} \sum_{v \in \mathcal{V}_m} \nabla_{\mathbf{W}} \mathcal{L}_m^{\text{Local}}(\mathbf{h}_v^{(L)}) - \frac{1}{M} \sum_{m=1}^{M} \frac{1}{|\mathcal{V}_m|} \sum_{v \in \mathcal{V}_m} \nabla_{\mathbf{W}} \mathcal{L}_m^{\text{Local}}(\mathbf{h}_v^{*(L)}) \right\|_2 \\
&\leq \frac{1}{M} \sum_{m=1}^{M} \left\| \frac{1}{|\mathcal{V}_m|} \sum_{v \in \mathcal{V}_m} \nabla_{\mathbf{W}} \mathcal{L}_m^{\text{Local}}(\mathbf{h}_v^{(L)}) - \frac{1}{|\mathcal{V}_m|} \sum_{v \in \mathcal{V}_m} \nabla_{\mathbf{W}} \mathcal{L}_m^{\text{Local}}(\mathbf{h}_v^{*(L)}) \right\|_2 \\
&\leq \frac{1}{M} \sum_{m=1}^{M} \frac{1}{|\mathcal{V}_m|} \sum_{v \in \mathcal{V}_m} \left\| \nabla_{\mathbf{W}} \mathcal{L}_m^{\text{Local}}(\mathbf{h}_v^{(L)}) - \nabla_{\mathbf{W}} \mathcal{L}_m^{\text{Local}}(\mathbf{h}_v^{*(L)}) \right\|_2 \\
&\leq \frac{1}{M} \sum_{m=1}^{M} \tau \cdot \sum_{\ell=1}^{L-1} \epsilon^{(\ell)} r_1^{L-\ell} r_2^{L-\ell} |\Delta \mathcal{G}_m|^{L-\ell},
\end{aligned}
\tag{14}
$$

which finishes the proof. $\qquad\square$

### A.5.2 Proof of Theorem 2

In this section, we prove the convergence of DIGEST under the synchronous setting. First, we introduce some notions, definitions and necessary assumptions.

**Preliminaries.** We consider GCN in our proof without loss of generality. We denote the input graph as $\mathcal{G} = (\mathcal{V}, \mathcal{E})$, $L$-layer GNN as $f$, feature matrix as $X$, weight matrix as $W$. The forward propagation of one layer of GCN is

$$
Z^{(\ell+1)} = P H^{(\ell)} W^{(\ell)}, \quad H^{(\ell+1)} = \sigma(Z^{(\ell)})
\tag{15}
$$

where $\ell$ is the layer index, $\sigma$ is the activation function, and $P$ is the propagation matrix following the definition of GCN (Kipf & Welling, 2016). Notice $H^{(0)} = X$. We can further define the $(\ell+1)$-th layer of GCN as:

$$
f^{(\ell+1)}(H^{(\ell)}, W^{(\ell)}) := \sigma(P H^{(\ell)} W^{(\ell)})
\tag{16}
$$

The backward propagation of GCN can be expressed as follow:

$$
G_H^{(\ell)} = \nabla_H f^{(\ell+1)}(H^{(\ell)}, W^{(\ell)}, G_H^{(\ell+1)}) := P^{\intercal} D^{(\ell+1)} (W^{(\ell+1)})^{\intercal}
\tag{17}
$$

$$
G_W^{(\ell+1)} = \nabla_W f^{(\ell+1)}(H^{(\ell+1)}, W^{(\ell)}, G_H^{(\ell+1)}) := (P H^{(\ell)})^{\intercal} D^{(\ell+1)}
\tag{18}
$$

where

$$
D^{(\ell+1)} = G_H^{(\ell)} \circ \sigma'(P H^{(\ell)} W^{(\ell+1)})
\tag{19}
$$

and $\circ$ represents the Hadamard product.

Under a distributed training setting, for each subgraph $\mathcal{G}_m = (\mathcal{V}_m, \mathcal{E}_m)$, $m = 1.2, \cdots, M$, the propagation matrix can be decomposed into two independent matrices, i.e. $P = P_{m,in} + P_{m,out}$, where $P_{m,in}$ denotes the propagation matrix for nodes inside the subgraph $\mathcal{G}_m$ while $P_{m,out}$ denotes that for neighbor nodes outside $\mathcal{G}_m$. If it will not cause confusion, we will use $P_{in}$ and $P_{out}$ in our future proof for simpler notation.

For DIGEST, the forward propagation of a single layer of GCN can be expressed as

$$
\begin{aligned}
\tilde{Z}_m^{(t,\ell+1)} &= P_{in} \tilde{H}_m^{(t,\ell)} \tilde{W}_m^{(t,\ell)} + P_{out} \tilde{H}_m^{(t-1,\ell)} \tilde{W}_m^{(t,\ell)} \\
\tilde{H}_m^{(t,\ell+1)} &= \sigma(\tilde{Z}_m^{(t,\ell)})
\end{aligned}
\tag{20}
$$

where we use $\tilde{H}$ to differentiate with the counterpart without staleness, i.e., $H$ (same for other variables). $t$ is the training iteration index. Similarly, we can define each layer as a single function

$$
\tilde{f}_m^{(t,\ell+1)}(\tilde{H}_m^{(t,\ell)}, \tilde{W}_m^{(t,\ell)}) := \sigma(P_{in} \tilde{H}_m^{(t,\ell)} \tilde{W}_m^{(t,\ell)} + P_{out} \tilde{H}_m^{(t-1,\ell)} \tilde{W}_m^{(t,\ell)})
\tag{21}
$$

Note that $\tilde{H}_m^{(t-1,\ell-1)}$ is not part of the input since it is the stale results from the previous iteration, i.e., it can be regarded as a constant in the current iteration.

Now we can give the definition of back-propagation in DIGEST:

$$\tilde{G}_{H,m}^{(t,\ell)} = \nabla_H \tilde{f}_m^{(t,\ell+1)}(\tilde{H}_m^{(\ell)}, \tilde{W}^{(\ell)}, \tilde{G}_{H,m}^{(\ell+1)})$$
$$:= P_{in}^{\mathsf{T}} \tilde{D}_m^{(t,\ell+1)}(\tilde{W}_m^{(t,\ell+1)})^{\mathsf{T}} + P_{out}^{\mathsf{T}} \tilde{D}_m^{(t-1,\ell+1)}(\tilde{W}_m^{(t,\ell+1)})^{\mathsf{T}} \tag{22}$$

$$\tilde{G}_{W,m}^{(t,\ell+1)} = \nabla_W \tilde{f}_m^{(t,\ell+1)}(\tilde{H}_m^{(t,\ell+1)}, \tilde{W}_m^{(t,\ell)}, \tilde{G}_{H,m}^{(t,\ell+1)})$$
$$:= (P_{in}\tilde{H}_m^{(t,\ell)} + P_{out}\tilde{H}_m^{(t-1,\ell-1)})^{\mathsf{T}} \tilde{D}_m^{(t,\ell+1)} \tag{23}$$

where

$$\tilde{D}_m^{(t,\ell+1)} = G_{H,m}^{(\ell)} \circ \sigma'(P_{in}\tilde{H}_m^{(t,\ell)}\tilde{W}_m^{(t,\ell)} + P_{out}\tilde{H}_m^{(t-1,\ell-1)}\tilde{W}_m^{(t,\ell)}) \tag{24}$$

In our proof, we use $\mathcal{L}(W^{(t)})$ to denote the global loss with GCN parameter $W$ after $t$ iterations, and use $\tilde{\mathcal{L}}_m(W_m^{(t)})$ to denotes the local loss for the $m$-th subgraph with model parameter $W_m^{(t)}$ after $t$ iterations computed by DIGEST.

**Assumptions.** Here we introduce some assumptions about the GCN model and the original input graph. These assumptions are standard ones that are also used in (Chen et al., 2018; Cong et al., 2021; Wan et al., 2022).

**Assumption 5.** *The loss function $Loss(\cdot, \cdot)$ is $C_{Loss}$-Lipchitz continuous and $L_{Loss}$-Lipschitz smooth with respect to the last layer's node representation, i.e.,*

$$|Loss(\mathbf{h}_v^{(L)}, \mathbf{y}_v) - Loss(\mathbf{h}_w^{(L)}, \mathbf{y}_v)| \leq C_{Loss}\|\mathbf{h}_v^{(L)} - \mathbf{h}_w^{(L)}\|_2 \tag{25}$$

*and*

$$\|\nabla Loss(\mathbf{h}_v^{(L)}, \mathbf{y}_v) - \nabla Loss(\mathbf{h}_w^{(L)}, \mathbf{y}_v)\|_2 \leq L_{Loss}\|\mathbf{h}_v^{(L)} - \mathbf{h}_w^{(L)}\|_2 \tag{26}$$

**Assumption 6.** *The activation function $\sigma(\cdot)$ is $C_\sigma$-Lipchitz continuous and $L_\sigma$-Lipschitz smooth, i.e.*

$$\|\sigma(Z_1^{(\ell)}) - \sigma(Z_2^{(\ell)})\|_2 \leq C_\sigma\|(Z_1^{(\ell)} - Z_2^{(\ell)})\|_2 \quad and \quad \|\sigma'(Z_1^{(\ell)}) - \sigma'(Z_2^{(\ell)})\|_2 \leq L_\sigma\|(Z_1^{(\ell)} - Z_2^{(\ell)})\|_2 \tag{27}$$

**Assumption 7.** $\forall \ell$ *that* $\ell = 1, 2, \cdots, L$*, we have*

$$\|W^{(\ell)}\|_F \leq K_W, \ \|P^{(\ell)}\|_P \leq K_W, \ \|X^{(\ell)}\|_F \leq K_X. \tag{28}$$

Now we can introduce the proof of our Theorem 2. We consider a GCN with $L$ layers that is $L_f$-Lipschitz smooth, i.e., $\|\nabla\mathcal{L}(W_1) - \nabla\mathcal{L}(W_2)\|_2 \leq L_f\|W_1 - W_2\|_2$.

**Theorem 5** (Formal version of Theorem 2)**.** *There exists a constant $E$ such that for any arbitrarily small constant $\epsilon > 0$, we can choose a learning rate $\eta = \frac{\sqrt{M}\epsilon}{E}$ and number of training iterations $T = (\mathcal{L}(W^{(1)}) - \mathcal{L}(W^*))\frac{E}{\sqrt{M}}\epsilon^{-\frac{3}{2}}$, such that*

$$\frac{1}{T}\sum_{t=1}^T \|\nabla\mathcal{L}(W^{(t)})\|^2 \leq \mathcal{O}(\frac{1}{T^{\frac{2}{3}}M^{\frac{1}{3}}}) \tag{29}$$

*where $W^{(t)}$ and $W^*$ denotes the parameters at iteration $t$ and the optimal one, respectively.*

*Proof.* Beginning from the assumption of smoothness of loss function,

$$\mathcal{L}(W^{t+1}) \leq \mathcal{L}(W^t) + \left\langle \nabla\mathcal{L}(W^t), W^{(t+1)} - W^{(t)} \right\rangle + \frac{L_f}{2}\|W^{(t+1)} - W^{(t)}\|_2^2 \tag{30}$$

Recall that the update rule of DIGEST is

$$W^{(t+1)} = W^{(t)} - \frac{\eta}{M}\sum_{m=1}^M \nabla\tilde{\mathcal{L}}_m(W_m^{(t)}) \tag{31}$$

so we have

$$
\mathcal{L}(W^t) + \left\langle \nabla\mathcal{L}(W^t), W^{(t+1)} - W^{(t)} \right\rangle + \frac{L_f}{2}\|W^{(t+1)} - W^{(t)}\|_2^2
$$
$$
=\mathcal{L}(W^t) - \eta \left\langle \nabla\mathcal{L}(W^t), \frac{1}{M}\sum_{m=1}^{M}\nabla\tilde{\mathcal{L}}_m(W_m^{(t)}) \right\rangle + \frac{\eta^2 L_f}{2}\left\|\frac{1}{M}\sum_{m=1}^{M}\nabla\tilde{\mathcal{L}}_m(W_m^{(t)})\right\|_2^2 \tag{32}
$$

Denote $\delta_m^{(t)} = \nabla\tilde{\mathcal{L}}_m(W_m^{(t)}) - \nabla\mathcal{L}_m(W_m^{(t)})$, we have

$$
\mathcal{L}(W^{t+1}) \leq \mathcal{L}(W^t) - \eta\left\langle \nabla\mathcal{L}(W^t), \frac{1}{M}\sum_{m=1}^{M}\left(\nabla\mathcal{L}_m(W_m^{(t)}) + \delta_m^{(t)}\right)\right\rangle
$$
$$
+ \frac{\eta^2 L_f}{2}\left\|\frac{1}{M}\sum_{m=1}^{M}\left(\nabla\mathcal{L}_m(W_m^{(t)}) + \delta_m^{(t)}\right)\right\|_2^2 \tag{33}
$$

Without loss of generality, assume the original graph can be divided evenly into M subgraphs and denote $N = |\mathcal{V}|$ as the original graph size, i.e., $N = M \cdot S$, where $S$ is each subgraph size. Notice that

$$
\nabla\mathcal{L}(W^t) = \frac{1}{N}\sum_{i=1}^{N}\nabla Loss(f_i^{(L)}, y_i) = \frac{1}{M}\left\{\sum_{m=1}^{M}\frac{1}{S}\sum_{i=1}^{S}\nabla Loss(f_{m,i}^{(L)}, y_{m,i})\right\} \tag{34}
$$

which is essentially

$$
\nabla\mathcal{L}(W^t) = \frac{1}{M}\sum_{m=1}^{M}\nabla\mathcal{L}_m(W_m^{(t)}) \tag{35}
$$

Plugging the equation above into Eq. 33, we have

$$
\mathcal{L}(W^{t+1}) \leq \mathcal{L}(W^t) - \frac{\eta}{2}\|\nabla\mathcal{L}(W^t)\|_2^2 + \frac{\eta^2 L_f}{2}\left\|\frac{1}{M}\sum_{m=1}^{M}\delta_m^{(t)}\right\|_2^2 \tag{36}
$$

which after rearranging the terms leads to

$$
\|\nabla\mathcal{L}(W^t)\|_2^2 \leq \frac{2}{\eta}(\mathcal{L}(W^t) - \mathcal{L}(W^{t+1})) + \eta L_f\left\|\frac{1}{M}\sum_{m=1}^{M}\delta_m^{(t)}\right\|_2^2 \tag{37}
$$

By taking $\eta < 1/L_f$, using the three assumptions defined earlier and Corollary A.10 in Wan et al. (2022), and summing up the inequality above over all iterations, i.e., $t = 1, 2, \cdots, T$, we have

$$
\frac{1}{T}\sum_{t=1}^{T}\|\nabla\mathcal{L}(W^{(t)})\|^2 \leq \frac{2}{\eta T}\left(\mathcal{L}(W^1) - \mathcal{L}(W^{T+1})\right) + \frac{\eta^2 E^2}{M}
$$
$$
\leq \frac{2}{\eta T}\left(\mathcal{L}(W^1) - \mathcal{L}(W^*)\right) + \frac{\eta^2 E^2}{M} \tag{38}
$$

where $W^*$ denotes the minima of the loss function and $E$ is a constant depends on $E'$.

Finally, taking $\eta = \frac{\sqrt{M}\epsilon}{E}$ and $T = (\mathcal{L}(W^{(1)}) - \mathcal{L}(W^*))\frac{E}{\sqrt{M}}\epsilon^{-\frac{3}{2}}$ finishes the proof.

$\square$

### A.5.3 PROOF OF THEOREM 3

By following Li et al. (2020); Chen et al. (2020); Chai et al. (2021) we make the assumption as below:

**Assumption 8.** $\forall\ m \in [M]$, $\|\nabla\tilde{\mathcal{L}}_m(W)\|_2 \leq V \cdot \|\nabla\mathcal{L}(W)\|_2$, and $\left\langle \nabla\mathcal{L}(W), \nabla\tilde{\mathcal{L}}_m(W)\right\rangle \geq \beta \cdot \|\nabla\mathcal{L}(W)\|_2^2$, where $V$ and $\beta$ are positive real numbers, i.e., $V, \beta \in \mathbb{R}^+$.

We naturally assume that each local copy of the global GCN model is also $L_f$-Lipschitz smooth, i.e., $\|\nabla\tilde{\mathcal{L}}_m(W_1) - \nabla\tilde{\mathcal{L}}_m(W_2)\|_2 \leq L_f\|W_1 - W_2\|_2, \forall\, m = 1, 2, \cdots, M$.

Now we can give the proof of Theorem 3.

**Theorem 6** (Formal version of Theorem 3). *Assume the global model $\mathcal{L}(W)$ is $C_f$-Lipschitz continuous and the delay is bounded, i.e., $\tau < K$. Further, assume the constants defined in Assumption 8 satisfy $\beta - \frac{V^2}{2} > 0$. Then, after $T$ global iterations on the server, asynchronous DIGEST converges to the optimal parameter $W^*$ by*

$$\frac{1}{T}\sum_{t=1}^{T}\|\nabla\mathcal{L}(W^{(t)})\|_2^2 \leq \frac{1}{\eta TB}\Big(\mathcal{L}(W^{(1)}) - \mathcal{L}(W^{(*)})\Big) + \frac{P(\eta)}{B}, \tag{39}$$

*where $B = \beta - \frac{V^2}{2}$ and $P(\eta) = \frac{1}{2}\eta^2 K^2 C_f^2 L_f^2 + (1+V)\eta KC_f^2 L_f$.*

*Proof.* By the smoothness assumption of global model,

$$\mathcal{L}(W^{(t+1)}) \leq \mathcal{L}(W^{(t)}) + \Big\langle\nabla\mathcal{L}(W^{(t)}), W^{(t+1)} - W^{(t)}\Big\rangle + \frac{L_f}{2}\|W^{(t+1)} - W^{(t)}\|_2^2. \tag{40}$$

Suppose at global iteration $t+1$, the server receives an update from subgraph $m$, where $m$ could be any value from 1 up to $M$. Then,

$$\mathcal{L}(W^{(t+1)}) \leq \mathcal{L}(W^{(t)}) - \eta\Big\langle\nabla\mathcal{L}(W^{(t)}), \nabla\tilde{\mathcal{L}}_m(W^{(t-\tau)})\Big\rangle + \frac{\eta^2 L_f}{2}\|\nabla\tilde{\mathcal{L}}_m(W^{(t-\tau)})\|_2^2, \tag{41}$$

where $\tau$ is the delay for subgraph $m$ when sending server its update.

Denote $r_m := \nabla\tilde{\mathcal{L}}_m(W^{(t-\tau)}) - \nabla\tilde{\mathcal{L}}_m(W^{(t)})$. Since $\tau \leq K$, by the Lipschitz smoothness of $\tilde{\mathcal{L}}_m$,

$$\begin{aligned}
\|r_m\|_2 &= \|\nabla\tilde{\mathcal{L}}_m(W^{(t-\tau)}) - \nabla\tilde{\mathcal{L}}_m(W^{(t)})\|_2 \\
&\leq L_f \cdot \|W^{(t-\tau)} - W^{(t)}\|_2 \\
&= L_f \cdot \|\sum_{i=t-\tau}^{t}\eta g_i\|_2 \\
&\leq \eta KL_f C_f,
\end{aligned} \tag{42}$$

where $g_i$ is the gradient or update the global server receives at iteration $i$.

Therefore,

$$\begin{aligned}
&\mathcal{L}(W^{(t+1)}) - \mathcal{L}(W^{(t)}) \\
&\leq -\eta\Big\langle\nabla\mathcal{L}(W^{(t)}), \nabla\tilde{\mathcal{L}}_m(W^{(t)}) + r_m\Big\rangle + \frac{\eta^2 L_f}{2}\|\nabla\tilde{\mathcal{L}}_m(W^{(t)}) + r_m\|_2^2 \\
&= \underbrace{-\eta\Big\langle\nabla\mathcal{L}(W^{(t)}), \nabla\tilde{\mathcal{L}}_m(W^{(t)})\Big\rangle}_{\text{I}} + \underbrace{\frac{\eta^2 L_f}{2}\|\nabla\tilde{\mathcal{L}}_m(W^{(t)})\|_2^2}_{\text{II}} + \underbrace{\frac{\eta^2 L_f}{2}\|r_m\|_2^2}_{\text{III}} \\
&\quad \underbrace{-\eta\Big\langle\nabla\mathcal{L}(W^{(t)}), r_m\Big\rangle}_{\text{IV}} + \underbrace{\eta^2 L_f\Big\langle\nabla\tilde{\mathcal{L}}_m(W^{(t)}), r_m\Big\rangle}_{\text{V}}
\end{aligned} \tag{43}$$

Now we want to find bounds for (I - V) above.

By Assumption 8, we have

$$\text{(I)} \leq -\eta\beta\|\nabla\mathcal{L}(\mathbf{W}^{(t)})\|_2^2, \tag{44}$$

and

$$\text{(II)} \leq \frac{1}{2}\eta^2 L_f V^2\|\nabla\mathcal{L}(\mathbf{W}^{(t)})\|_2^2. \tag{45}$$

By our previous result on $\|r_m\|_2$, we have

$$\text{(III)} \leq \frac{1}{2}\eta^3 K^2 L_f^2 C_f^2 \tag{46}$$

Taking $\eta \leq 1/L_f$, we have

$$\text{(IV)} + \text{(V)} \leq \eta \left\langle \nabla \tilde{\mathcal{L}}_m(W^{(t)}) - \nabla \mathcal{L}(W^{(t)}), r_m \right\rangle \tag{47}$$

By Cauchy-Schwartz inequality, triangle inequality and Assumption 8,

$$
\begin{aligned}
\text{(IV)} + \text{(V)} &\leq \eta \|\nabla \tilde{\mathcal{L}}_m(W^{(t)}) - \nabla \mathcal{L}(W^{(t)})\|_2 \cdot \|r_m\|_2 \\
&\leq \eta^2 K C_f L_f \cdot \|\nabla \tilde{\mathcal{L}}_m(W^{(t)}) - \nabla \mathcal{L}(W^{(t)})\|_2 \\
&\leq \eta^2 K C_f L_f \cdot \left( \|\nabla \tilde{\mathcal{L}}_m(W^{(t)})\|_2 + \|\nabla \mathcal{L}(W^{(t)})\|_2 \right) \\
&\leq (1+V)\eta^2 K C_f L_f \cdot \|\nabla \mathcal{L}(W^{(t)})\|_2 \\
&\leq (1+V)\eta^2 K C_f^2 L_f
\end{aligned}
\tag{48}
$$

Put everything together, we have

$$\mathcal{L}(W^{(t+1)}) - \mathcal{L}(W^{(t)}) \leq \left( \frac{1}{2}\eta V^2 - \eta\beta \right) \cdot \|\nabla \mathcal{L}(W^{(t)})\|_2^2 + \frac{1}{2}\eta^3 K^2 L_f^2 C_f^2 + (1+V)\eta^2 K C_f^2 L_f. \tag{49}$$

Hence,

$$
\begin{aligned}
\|\nabla \mathcal{L}(W^{(t)})\|_2^2 &\leq \left( \eta\beta - \frac{1}{2}\eta V^2 \right)^{-1} \cdot \left( \mathcal{L}(W^{(t)}) - \mathcal{L}(W^{(t+1)}) \right) \\
&\quad + \left( \eta\beta - \frac{1}{2}\eta V^2 \right)^{-1} \cdot \left( \frac{1}{2}\eta^3 K^2 L_f^2 C_f^2 + (1+V)\eta^2 K C_f^2 L_f \right).
\end{aligned}
\tag{50}
$$

Summing up from $t = 1$ to $T$ and taking the average,

$$
\begin{aligned}
\frac{1}{T} \sum_{t=1}^{T} \|\nabla \mathcal{L}(W^{(t)})\|_2^2 &\leq \frac{1}{\left( \eta\beta - \frac{1}{2}\eta V^2 \right) T} \left( \mathcal{L}(W^{(1)}) - \mathcal{L}(W^{(*)}) \right) \\
&\quad + \left( \eta\beta - \frac{1}{2}\eta V^2 \right)^{-1} \cdot \left( \frac{1}{2}\eta^3 K^2 L_f^2 C_f^2 + (1+V)\eta^2 K C_f^2 L_f \right) \\
&\leq \frac{1}{\eta T B} \left( \mathcal{L}(W^{(1)}) - \mathcal{L}(W^{(*)}) \right) + \frac{P(\eta)}{B},
\end{aligned}
\tag{51}
$$

where $B = \beta - \frac{V^2}{2}$ and $P(\eta) = \frac{1}{2}\eta^2 K^2 C_f^2 L_f^2 + (1+V)\eta K C_f^2 L_f$. $\qquad \square$

