# OpenReview forum: "Distributed Graph Neural Network Training with Periodic Stale Representation Synchronization"
_ICLR.cc/2023/Conference — Submitted to ICLR 2023_

### Official Review · Reviewer_2my9 · 2022-10-23

**Confidence:** 3
**Correctness:** 3
**Technical Novelty And Significance:** 3
**Empirical Novelty And Significance:** 3
**Recommendation:** 6

**Clarity, Quality, Novelty And Reproducibility:**

This paper presents a distributed GNN training framework very clearly. Both the theoretical analysis and the experiments are convincing. Though there are some works, e.g., [1], that share similar ideas to train GNNs with stale representation, this paper has its contribution as a system implementation with stale representation synchronization and pipelined I/O.

[1] Stochastic Training of Graph Convolutional Networks with Variance Reduction, Jianfei Chen, Jun Zhu, Le Song, ICML 2018


**Strength And Weaknesses:**

Strengths:
1. The distributed training of GNNs is an important problem. The motivation of this paper is clear and the proposed method is simple yet effective.
2. The experiments show the speedup and scalability are good.
3. This paper gives theoretical bound and convergence analysis of the proposed training method.

Concerns:
1. A question is, if the I/O is pipelined and hidden as in Figure 2, why is it still needed to sync **periodically**(as we should be able to sync in each epoch without much addition time cost)? And as shown in Figure 6, is there any way to determine or suggest the best synchronization frequency?
2. This method proposed in this paper shows nice scalability in a single-machine multi-GPU environment. It could be more interesting if it could be extended to a multi-machines environment with nice scalability.

**Summary Of The Paper:**

This paper proposes a method to achieve a trade-off between information loss(partition-based) and communication cost(propagation-based) distributed GNN training. The key design is to synchronize representation to an in-memory KV store periodically with pipelined IO, so that the GNN training is aware of the full graph and without frequent communication overhead.

**Summary Of The Review:**

This paper proposes a distributed GNN training framework by balancing partition-based and propagation-based distributed GNN training. The key designs are periodic stale representation synchronization with pipelined I/O. The results are convincing and I think the merits outweigh the flaws.

---

> ### Author Response · Authors · 2022-11-19
> **Author Response to Reviewer 2my9**
>
> Dear Reviewer,
>
> We are happy to see that you found our method interesting and sound. To better address your questions, we have provided more details and explanations **in this rebuttal directly**. We will incorporate these updates in the final revision of the paper. Please refer to our response below for detail.
>
> >*“A question is, if the I/O is pipelined and hidden as in Figure 2, why is it still needed to sync **periodically** (as we should be able to sync in each epoch without much addition time cost)? And as shown in Figure 6, is there any way to determine or suggest the best synchronization frequency?”*
>
> **A1:** Thanks for catching that. In fact, we found in our experimental evaluation that the latency of PULL and PUSH I/O operations cannot be fully overlapped with the concurrent epoch training. According to our measurement, it takes around 1-2 seconds for DIGEST to PULL or PUSH the representations, the aggregate cost of which is at the same order as, or slightly higher than, the per-epoch training time. (Optimizing the I/O latency is part of our future work.)
> Therefore, even though concurrently overlapping I/O and compute could improve the overall training time to some extent, DIGEST still needs to tackle the tradeoff between training time, F1 score, and the I/O overhead. Figure 6 studied this tradeoff: with relatively large synchronization intervals (lower synchronization frequencies), DIGEST saw reduced training time at the expense of slightly impacted F1 score. Relatively smaller synchronization intervals would lead to higher F1-score performance but longer training time. Based on our experience, denser graphs typically need a higher synchronization frequency than sparer graphs, due to a high probability of having out-of-subgraph nodes. We suggest that empirically choosing an interval value that is not too small nor too big (e.g., $10$) may lead to balanced training time vs. F1 scores.
>
> >*“This method proposed in this paper shows nice scalability in a single-machine multi-GPU environment. It could be more interesting if it could be extended to a multi-machines environment with nice scalability.”*
>
> **A2:** DIGEST in a multi-machines environment can get similar but slight lower scalability as in a single-machine, multi-GPU environment, due to the network communication cost introduced in a multi-machine distributed setup.
> That is, between the two training environments, the only difference is the communication bandwidth among subgraphs. Assume that the communication time for a single subgraph in a single-machine, multi-GPU environment is $t_1$ and the communication time for a single subgraph in multi-machines environment is $t_2$ (which is larger than $t_1$). $t_2 - t_1$ can be bounded to a small-enough value by using high-bandwidth, low-latency connectivity between machines, e.g., the 25 Gigabit Ethernet (GbE) or 100 GbE connectivity is not uncommon in today's GPU clusters. Since the stale representations do not need to be synchronized every epoch, such difference can be further reduced. We have added new experiments on an $N$-machine cluster, where $N$ is equal to the total number of GPUs used in the test and each machine is configured with one GPU. We present the results of the $N$-machine cluster in the second row of the table below. We see that in the multi-machine environment DIGEST still achieves on-par scalability as that of a multi-GPU, single-machine setup (first row of the table).
>
> | # of GPUs                 | 4             | 5             | 6             | 7             |
> |---------------------------|---------------|---------------|---------------|---------------|
> | Multi-GPU, single-machine | 11.5$\times$  | 13.55$\times$ | 14.5$\times$  | 16.12$\times$ |
> | Single-GPU, multi-machine | 10.35$\times$ | 11.97$\times$ | 12.71$\times$ | 13.94$\times$ |

---

### Official Review · Reviewer_UPkq · 2022-10-23

**Confidence:** 3
**Clarity, Quality, Novelty And Reproducibility:** The paper is well written. The idea i…
**Correctness:** 3
**Technical Novelty And Significance:** 2
**Empirical Novelty And Significance:** 2
**Recommendation:** 5

**Strength And Weaknesses:**

## Strength:
1. The GNN training framework proposed in this paper drops no edges while avoiding communication overhead, showing both good performances of accuracy and speedup.

2. The periodic stale representation synchronization technique using periodically synchronous pull/push operations for the representations is novel and able to approximate the representations of nodes.

3. The authors provide detailed proof of performance and convergence for both synchronous and asynchronous versions.

## Weakness:
1. Does KVS stores the representations of all the nodes on a graph? It would be better if the authors could provide the memory consumption of different methods. If the memory consumption of DIGEST is large, it may be hard for it to handle as large graphs as both partition-based and propagation-based methods.

2. The experiments should use larger datasets, such as ogbn-papers100M and mag-240. The datasets used in this paper can be easily saved on a single machine. (The data size of the largest dataset ogbn-products is only 2.8G.)

**Summary Of The Paper:**

This paper explores the trade-off between partition-based and propagation-based distributed training methods, taking advantage of both low communication overhead and no information loss. To do this, it saves a subgraph on each device and allows each device utilizes the stale representation of the neighbors of the subgraph saved on other devices. The results show more than 20 times speedup compared with the state-of-the-art distributed GNN training frameworks. This paper also proves the convergence of the proposed method.

**Summary Of The Review:**

The paper is well written. The proposed technique is intuitive, but not very impressive.  The experiments are insufficient.

---

> ### Author Response · Authors · 2022-11-19
> **Author Response to Reviewer UPkq**
>
> Dear Reviewer,
>
> Thank you for your appreciation of the novelty of our method and the quality of our presentation. To better address your questions, we have provided more details and explanations **in this rebuttal directly and in our re-uploaded paper revision**. We will also incorporate these updates in the final revision of the paper. Please refer to our response below for detail.
>
> >*“Does KVS stores the representations of all the nodes on a graph? It would be better if the authors could provide the memory consumption of different methods. If the memory consumption of DIGEST is large, it may be hard for it to handle as large graphs as both partition-based and propagation-based methods.”*
>
> **A1:** The KVS is responsible for storing the representations of all the nodes in a graph. The representations are stored in the memory of the host server instead of the GPUs, the latter of which is rather limited. We implemented the in-memory KVS with Apache Plasma, which is a shared memory storage that supports efficient, shared-memory-based inter-process communication (IPC) for multiple training processes located on the same server. However, extending our current KVS implementation to a fully-distributed storage system is trivial.
> Using off-the-shelf, high-performance distributed in-memory KVSes such as Redis is one option. Alternatively, we could also implement a simple client library, which can be used by the training process for key-value item mapping (e.g., using the commonly-used consistent hashing algorithm) and remote representation retrieval/storage, and with the client library, we could deploy a cluster of Plasma storage processes either on a dedicated storage cluster or on the same training server cluster to support distributed representation storage.
> The overall memory consumption required to store representation data can be calculated with the following equation:
> \begin{equation}
> KVS\ memory\ usage = (L-1) \times dim \times |V| \times s
> \end{equation}
>
> where $L$ is the total number of layers of the model, $dim$ denotes the hidden dimension, $|V|$ represents the number of nodes in the graph, and $s$ is the size of the data type in Python NumPy.
> For the $float32$ data type, it takes $4$ bytes for every single value. With the provided formula, for a 3-layer GNN model, training large graph dataset such as OGB-Products (with $2,449,029$ nodes and $128$ hidden dimensions), the extra host memory consumption for the representations is around $2 * 128 * 2449029 * 4 / 1024/ 1024/ 1024 = 2.336$ GB.
> DIGEST exhibits an interesting tradeoff: it uses a small amount of extra memory overhead for storing stale representations to enable the disaggregation of the compute and storage for higher scalability and more flexibility. We also argue that a small host memory cost of several GBs is negligible considering today's multi-GPU servers are equipped with hundreds of GBs if not more than a few TBs of host memory. For example, AWS EC2's p3.16xlarge is equipped with 8 Nvidia Tesla V100 GPUs with 488 GBs of host memory: https://aws.amazon.com/blogs/aws/new-amazon-ec2-instances-with-up-to-8-nvidia-tesla-v100-gpus-p3/}.
>
> For the concern of GPU memory consumption, we compare DIGEST with GraphSAGE, LLCG, and DGL by including all the information inside a GNN's receptive field in a single optimization step. The comparison results below show that DIGEST has the lowest GPU memory consumption across all four systems. We also reported these new results in the revised paper (Table 4).
> |   Model   | OGB-Arxiv | OGB-Products |
> |:---------:|:---------:|:------------:|
> | GraphSAGE |  0.40 GB  |    0.92 GB   |
> |    DGL    |  0.64 GB  |    1.78 GB   |
> |    LLCG   |  0.23 GB  |    0.36 GB   |
> |   DIGEST  |  0.22 GB  |    0.36 GB   |
>
> >*“The experiments should use larger datasets, such as ogbn-papers100M and mag-240. The datasets used in this paper can be easily saved on a single machine. (The data size of the largest dataset ogbn-products is only 2.8G.)”*
>
> **A2:** With more GPU resources, DIGEST can also be used to train extremely large graphs like OGB-papers100m and mag-240. We conducted an experiment on OGB-paper100m under the same setting used by PipeGCN [5], which consists of 32 GPUs. We report the training time per epoch in the following table. We also reported these newly-obtained results in the revised paper (Figure 11).
>
> | Model                                   | DGL | LLCG | PipeGCN | DIGEST |
> |:-----------------------------------------:|:-----:|:------:|:---------:|:--------:|
> | Time/Epoch (seconds) for OGB-papers100m |  ---  |   ---  |   7.1s  |  5.6s  |
>
> DGL and LLCG were not able to finish the training given our time budget of 200 seconds, therefore their results are omitted. We see that DIGEST reduces the training time per epoch by $21.13$% compared to PipeGCN.
>
> **References**
>
> [5] Wan, Cheng, et al. PipeGCN: Efficient full-graph training of graph convolutional networks with pipelined feature communication. 2022.

---

### Official Review · Reviewer_xBXJ · 2022-10-24

**Confidence:** 4
**Correctness:** 4
**Technical Novelty And Significance:** 3
**Empirical Novelty And Significance:** 3
**Recommendation:** 8

**Clarity, Quality, Novelty And Reproducibility:**

Quality: The proposed method is both theoretically and empirically sound to me. Extensive theoretical analyses are shown for convergence guarantee and stale gradients error bound.  Empirical studies are comprehensive and shows great speedup and performance of the proposed method.

Clarity: In general, this paper is well-written with great presentation. Also, I found this paper quite easy to follow.

Originality: As far as I know, this paper makes non-trivial contributions for distributed partition-parallel training of GNNs with stale representation, which could handle information loss and communication overhead simultaneously. Novel system design and extensive theoretical analyses further round up the good work.


**Details Of Ethics Concerns:**

No ethics concerns found in this paper.

**Strength And Weaknesses:**

Strengths:

1.	As far as I know, the originality of this paper is good. Considering stale representation under distributed training of GNNs is interesting and important.

2.	The proposed method is both theoretically and technically sound to me.

3.	This paper makes non-trivial contribution on system design and implementation, including both synchronous and asynchronous variants of the proposed method, and the key-value storage (KVS) for stale representations.

4.	The authors provided very extensive and novel theoretical analyses over the proposed methods. Convergence guarantees for both versions of DIGEST and the error bound of the approximated gradients due to the staleness are given with rigorous proof.

5.	The empirical results show a significant speedup of the proposed methods compared with other state-of-the-art distributed GNN methods on large real-world graph datasets.

Weaknesses:

1.	The authors are encouraged to better discuss the motivations and technical details of the asynchronous version of DIGEST. Also, how is the algorithm of asynchronous DIGEST different from that of synchronous one as shown in Algorithm 1 in appendix?

2.	Since graph partition algorithms can play an important role in distributed partition-based training of GNNs, I am curious how does different partition affect the performance of DIGEST?


**Summary Of The Paper:**

This paper considers the problem of distributed training of Graph Neural Networks (GNNs) over large-scale graphs. The authors proposed a framework called Distributed Graph Representation Synchronization (DIGEST), which leverages the stale representation of neighbors from other subgraphs to eliminate the information loss caused by dropped edges in partition-based methods and avoid communication overhead as in propagation-based methods. Specifically, the stale representations are stored in the central server and each subgraph communicate with the server to pull / push the stale representations during training. From system design perspective, the authors proposed both synchronous and asynchronous versions of DIGEST to further handle the straggler issue. Also, a key-value storage (KVS) design is introduced to implement the shared memory for the stale representations for better efficiency. Extensive theoretical analyses are provided, including convergence guarantee and the error bound of approximated gradients. Experimental results on several large-scale real-world graph datasets showed great performance and speedup of the proposed method compared with other distributed GNN methods.

**Summary Of The Review:**

In summary, this paper delivered a non-trivial and meaningful exploration of distributed partition-parallel training of GNNs. Both system design and theoretical analyses are novel and interesting. Empirical results are provided with great performance. Also, the paper presentation is good to me. Hence, I would recommend accept of this paper.

---

> ### Author Response · Authors · 2022-11-19
> **Author Response to Reviewer xBXJ**
>
> Dear Reviewer,
>
> We are happy to see that you found our method interesting and sound. To better answer your questions, we have provided more details and explanations **in this rebuttal directly**. We will incorporate these updates in the final revision of the paper. Please refer to our response below for detail.
>
> > *“The authors are encouraged to better discuss the motivations and technical details of the asynchronous version of DIGEST. Also, how is the algorithm of asynchronous DIGEST different from that of synchronous one as shown in Algorithm 1 in the appendix?”*
>
> **A1.** For the synchronous version, each subgraph will start the next training epoch until all the subgraphs finish their current training epoch. The synchronous training mode is not ideal when there is a load imbalance among all the involved subgraphs. There are two cases that may result in load imbalance: one case is when different subgraphs may require different training times, and the other is when the training environment consists of heterogeneous training devices. While the former case is not evaluated (however, it could happen in real-world graph applications), the latter case is common in real-world data centers: for example, most of today's data centers are equipped with GPU resources from different generations, which may vary in terms of computing power and memory capacity ([3,4]).
>
> When the training time is imbalanced, all the subgraphs have to wait for the slowest subgraph to finish, and this creates a training bottleneck with prolonged overall training time. To address this, we designed an asynchronous DIGEST, called DIGEST-A. In DIGEST-A, the training of each subgraph is independent of the other. With this asynchronous training approach, distributed subgraphs can benefit from the training results of other subgraphs immediately when a single subgraph finishes its current epoch, therefore significantly improving the overall training time.
>
> > *“Since graph partition algorithms can play an important role in distributed partition-based training of GNNs, I am curious how different partitions affect the performance of DIGEST?”*
>
> **A2.** For each node in a partitioned subgraph, METIS guarantees a high probability that its neighbors are located in the same subgraph. We chose to use METIS because METIS is widely used for large graph datasets in practice. or other graph partition algorithms such as RANDOM, we observe **similar** results, and we will report the details of various graph partitioning algorithms in the final version of the paper if needed.
>
> **References**
>
> [3] Jia, Xianyan, et al. Whale: Efficient Giant Model Training over Heterogeneous {GPUs}. In the Proceedings of 2022 USENIX Annual Technical Conference (USENIX ATC 22).
>
> [4] Park, Jay H., et al. HetPipe: Enabling Large {DNN} Training on (Whimpy) Heterogeneous {GPU} Clusters through Integration of Pipelined Model Parallelism and Data Parallelism. In the Proceedings of 2020 USENIX Annual Technical Conference (USENIX ATC 20).

---

### Official Review · Reviewer_yVeu · 2022-10-25

**Confidence:** 4
**Correctness:** 2
**Technical Novelty And Significance:** 3
**Empirical Novelty And Significance:** Not applicable
**Recommendation:** 5

**Clarity, Quality, Novelty And Reproducibility:**

# Clarity

The theoretical results of the paper need further revisions. Assumptions being made need to be clearly clarified in the main theorem. For instance, there are lots of assumptions being made in the proof but never mentioned in Theorem 2 and Theorem 3. Furthermore, the constants in the Theorem need to be explained, such as $E, M, P(\eta)$ in Theorem 2 and 3.

# Originality

The major idea of the paper follows GNNAutoScale, and the asynchronous update is inspired by extensive research on asynchronous distributed optimization. The theoretical analyses and assumptions closely follow PipeGNN [1]. Overall, the originality is not strong.






**Strength And Weaknesses:**

# Strength

1. The paper proposes a distributed GNN training framework that synergies the benefits of partition-based and propagation-based methods. The motivation is clearly demonstrated and the approach is valid.

2. Theoretical analyses demonstrate the impact of embedding error and staleness bound on prediction error and convergence behavior. This helps justify the impact of the staleness (but concerns are discussed below).

3. The experiments validate the effectiveness of the proposed algorithm, and the improvement over baselines is significant.

# Weakness

1. The major idea of the paper follows GNNAutoScale, and the asynchronous update is inspired by extensive research on asynchronous distributed optimization. The theoretical analyses and assumptions closely follow PipeGNN [1]. Overall, the originality is a bit weak.

2. The theoretical statements of the paper need further clarification. Assumptions being made need to be clearly clarified in the main theorem. For instance, there are lots of assumptions being made in the proof but they are never mentioned in Theorem 2 and Theorem 3. Furthermore, the constants in the Theorem need to be explained, such as $E, M, P(\eta)$ in Theorem 2 and 3.

3. There are multiple concerns with the theoretical analysis.

(1) First, the error presented in Theorem 1 can be potentially very large, which grows exponentially with the number of layers and has a bad dependency on the maximum node degree. A detailed empirical study of this approximation error will be helpful to clarify the impact of the error.

(2) Second, the backward propagation process for gradient computation neglects the gradient computation through the out-subgraph stale representations. In other words, each local model considers the node representation from other machines to be constant. The paper and the corresponding theorem do not take this into account. In fact, this will invalidate one important step in the theoretical proof Eq. (34): $\nabla L(W^t) = 1/M \sum_1^M \nabla L_m(W_m^t)$. This might cause major flaws in the proof and need to be addressed.

4. The algorithm is only compared with 2 baselines that do not represent the state-of-art algorithms. It is suggested to also compare with algorithms such as PipeGNN and sampling-based methods such as GNNAutoScale.

[1] PipeGCN: Efficient Full-Graph Training of Graph Convolutional Networks with Pipelined Feature Communication

**Summary Of The Paper:**

The paper introduces a distributed extension of GNNAutoScale (DIGEST) and proposes an asynchronous representation update mechanism (DIGEST-A) to reduce communication overhead for node embedding updates. Theoretical analyses such as forward error bound and convergence analysis are provided. Experiments show promising speedup in training time.

**Summary Of The Review:**

The paper introduces a simple and practical distributed GNN framework. Theoretical analyses are presented but concerns need to be addressed. Experiments demonstrate significant speedup but more baselines will be beneficial.

---

> ### Author Response · Authors · 2022-11-19
> **Author Response to Reviewer yVeu (Part 2)**
>
> > *“There are multiple concerns with the theoretical analysis. First, the error presented in Theorem 1 can be potentially very large, which grows exponentially with the number of layers and has a bad dependency on the maximum node degree. A detailed empirical study of this approximation error will be helpful to clarify the impact of the error.”*
>
> **A3.** Thank you for your suggestion. **We have added one more experiment** on the gradient approximation error to validate the impact of the error in practice. Please refer to **Appendix 3.3 on Pages 16 and 17** in our revised version for the updated plots. As can be seen in those figures, during the training phase the gradients calculated by DIGEST quickly converge to the ground-truth gradients typically within 10-15 epochs (the entire training converges after around 200 epochs). In other words, for the majority of training epochs, the error of gradients is very small and the impact could be negligible.
>
> > *“Second, the backward propagation process for gradient computation neglects the gradient computation through the out-subgraph stale representations. In other words, each local model considers the node representation from other machines to be constant. The paper and the corresponding theorem do not take this into account. In fact, this will invalidate one important step in the theoretical proof Eq.(34): $\nabla L(W^{(t)}) = (1/M)\sum_{m=1}^{M} \nabla L_{m}(W_{m}^{(t)})$. This might cause major flaws in the proof and need to be addressed.”*
>
> **A4.** This is a misunderstanding of our proposed method in the paper. The out-of-subgraph node representations are stale, i.e., **these representations were computed from the previous epoch** instead of being computed in the current one. That being said, the out-of-subgraph representations are **not** intermediate variables in the computation graph of the current epoch but rather some inputs or constants. As a result, our Eq.(34) is well-defined and theoretically sound.
>
> > *“The algorithm is only compared with 2 baselines that do not represent the state-of-art algorithms. It is suggested to also compare with algorithms such as PipeGNN and sampling-based methods such as GNNAutoScale.”*
>
> **A5.** Thanks for the suggestions. We have added the experiments on PipeGCN and GNNAutoScale, and also added new larger datasets for further strengthening the evaluation.
>
> Specifically, We trained on the OGB-Products dataset using DIGEST/DIGEST-A, PipeGCN, and GNNAutoScale in the same heterogeneous environment mentioned in Section 5.2. We report the training time taken to reach the target validation F1 score ($91$%) and time per epoch in the table below. Since there is no "epoch" concept in an asynchronous setting, the metric of time per epoch for DIGEST-A is omitted. We can see that DIGEST requires slightly higher time per epoch than GNNAutoScale but reduces the time per epoch by $24.13$% compared to PipeGCN. In the meantime, DIGEST-A achieves the lowest training time in order to reach the target F1 score and reduces the training time by $48.98$% and $19.12$% compared to PipeGCN and GNNAutoScale, respectively.
>
> | Model      | Training time to reach target Val. F1 score (91\%) | Time/Epoch      |
> | :---        |    :----:   |       :----:  |
> | PipeGCN      | 1795.9 s       | 15.87 s   |
> | GNNAutoScale   | 1132.7 s        | 11.7 s     |
> | DIGEST   | 2152 s        | 12.04 s      |
> | DIGEST-A   | 916.12  s     | ---     |
>
> To further demonstrate the efficiency of DIGEST, we evaluated DIGEST, PipeGCN, and GNNAutoScale on a large graph OGB-papers100m, which consists of $111$ million nodes and $1.6$ billion edges. The experiments were performed in a homogeneous environment with a total of 32 GPUs distributed across 4 EC2 VMs. Since mini-batches in GNNAutoScale are trained in a serial manner instead of a parallel distributed setting, only one GPU is used by GNNAutoScale. DIGEST reduces the time per epoch by $21.13$% compared to distributed  PipeGCN. The time per epoch results is reported in the Table below. We also reported these newly-obtained results in the revised paper (Figure 11).
>
> |   Model    | PipeGCN               |      GNNAutoScale             |        DIGEST     |
> | :---        |    :----:   |       :----:  |     :----:  |
> | **Time/Epoch (seconds) for OGB-papers100m**      | 7.1 s       | 124.4 s   |   5.6 s   |

---

> > ### Comment · Reviewer_yVeu · 2022-11-23
> > **Further comments**
> >
> > Dear authors,
> >
> > Thanks for your response but they do not resolve my concerns about the theoretical analysis.
> >
> > Regarding the gradient approximation error, how is the ground-truth gradient computed? Is it computed in a single machine or distributed setting as described in the paper? Does it compute the gradient through the state representation?
> >
> > Regarding the theoretical analysis, I do not misunderstand the idea. In fact, what is described in the rebuttal "the out-of-subgraph representations are not intermediate variables in the computation graph of the current epoch but rather some inputs or constants" is exactly what I pointed out in my initial comments "each local model considers the node representation from other machines to be constant".
> >
> > The predictions of one batch of data have a dependency on other batches but the dependency is simply ignored when computing the backpropagation. I do not think the finite-sum decomposition in Eq.(34) is well-defined and theoretically sound by ignoring the backpropagation through the state representations.

---

> > > ### Author Response · Authors · 2022-11-28
> > > **Author Response to Reviewer yVeu (2nd Round, Part 2)**
> > >
> > > Below are the definitions for some notations in our response part 1.
> > >
> > > $\\mathbf{H}\_{in}^{(\\ell,m)}$ and $\\mathbf{\tilde{H}}\_{out}^{(\\ell,m)}$ denotes the matrix of in-subgraph node representations and out-of-subgraph stale representations at $\\ell$-th layer on subgraph $\\mathcal{G}\_m$, respectively. $F$ denotes the forward propagation function of one layer of GNN for compact formula. $\\mathbf{P}\_{in}^{(m)}$ and $\\mathbf{P}\_{out}^{(m)}$ denotes the propagation matrix for in-subgraph nodes and out-of-subgraph nodes of $\\mathcal{G}\_m$, respectively, and we have $\\mathbf{P}\_m=\\mathbf{P}\_{in}^{(m)}+\\mathbf{P}\_{out}\^{(m)}$ where $\\mathbf{P}\_m$ is the original propagation matrix for subgraph $\\mathcal{G}\_m$.

---

> > > ### Author Response · Authors · 2022-11-28
> > > **Author Response to Reviewer yVeu (2nd Round, Part 1)**
> > >
> > > Dear Reviewer,
> > >
> > > Thank you for your timely feedback and question regarding our rebuttal responses. Please refer to our responses below for detail.
> > >
> > > > *“Regarding the gradient approximation error...?”*
> > >
> > > **A1.** The ground-truth gradient is computed in a distributed setting as stated in Eq.(10) without any stale representation. More specifically, for each epoch, subgraphs in the ground-truth approach receive real-time representations of out-of-subgraph nodes from corresponding subgraphs. We also compared the gradients of DIGEST with the gradients when training graphs in a full-batch manner with a single GPU, and observed similar error curves. We will modify section A.3.3 and add a detailed description regarding the ground-truth gradient to avoid confusion in the future.
> > >
> > > > *“Regarding the theoretical analysis...?”*
> > >
> > > **A2.** Thank you for your response which makes us clearer about your concern. We propose to answer your question from two perspectives:
> > >
> > > **(1)** **The dependency on other batches is not ignored and is completely considered, yet with some delay.** This way our method can achieve a more compelling trade-off between partition-based methods which ignore dependency on batches and propagation-based methods which considers dependency on batches without delay and hence suffer from high communication and low concurrency.
> > >
> > > **The gradient flow from other subgraphs still exists, though in an aggregated/pruned and stale way.** To see this, ALL out-of-subgraph node information has been aggregated in the stale representation of the 1-hop neighbors during the previous epoch. The aggregated stale representation, though constant from an optimization perspective in the next epoch, allows each subgraph to receive complete out-of-subgraph information and conduct the backpropagation in a (delayed) full-graph manner.
> > >
> > > **(2)** Under the trade-off and setting mentioned above, we would like to explain why Eq.(34) is well-defined and theoretically sound on the mathematical level. We believe our first point above is most related to your question, and our second point below is more like a re-explanation just in case anything was not clear enough. Our Eq.(34), i.e.,
> > >
> > > $$\\nabla \\mathcal{L}(W\^{(t)}) = \\frac{1}{M}\\sum\_{m=1}^{M} \\nabla \\mathcal{L}\_{m}(W\_{m}^{(t)}), \\quad t=1,2,\\cdots,T,$$
> > >
> > > states that the global gradient for the trained GNN is equivalent to the average of each local gradient. This equation holds trivially under i.i.d data (e.g., images) while in general does not hold under non-i.i.d data such as graphs since there are dependencies between different local subgraphs. However, in DIGEST, we adopt the approximation of out-of-subgraph nodes via stale representations, which **allows both forward propagation and backward propagation (thus the entire computation graph) on each local subgraph to be independent**. To see this, given a node $v\\in\\mathcal{G}\_{m}(\\mathcal{V}\_{m},\\mathcal{E}\_{m})$, where $\\mathcal{G}_{m}$ denotes the $m$-th subgraph, the forward propagation for the $(\\ell+1)$-th layer of DIGEST is achieved as
> > >
> > > $$\\mathbf{h}_{v}^{(\\ell+1)} =  \\Psi^{(\\ell+1)} \\bigg( \\mathbf{h}\_{v}^{(\\ell)},  \\Phi^{(\\ell+1)} \\Big( \\big\\{  \\mathbf{h}\^{(\\ell)}_u :  u \in \\mathcal{N}(v) \\cap \\mathcal{V}_m \\big\\} \cup   \\big\\{\\mathbf{\\tilde{h}}\^{(\\ell)}_u :  u \in \\mathcal{N}(v) \\setminus \\mathcal{V}_m \\big\\}   \Big)  \\bigg),$$
> > >
> > > where the out-of-subgraph node representations $\\mathbf{\\tilde{h}}\^{(\\ell)}\_u$ are computed from the previous epoch, thus the forward propagation of subgraph $\mathcal{G}_{m}$ is independent of those on other subgraphs. On the other hand, the backward propagation over a single layer of the GCN can be formulated as
> > >
> > > $$ \\quad \\frac{\\partial}{\\partial\\mathbf{W}\^{(\\ell+1)}_{m}} F\\Big(\\mathbf{H}\_{in}^{(\\ell,m)},\\mathbf{\\tilde{H}}\_{out}^{(\\ell,m)}\\Big) = \\frac{\\partial}{\\partial \\mathbf{W}\^{(\\ell+1)}\_{m}}\\sigma \\Big(\\mathbf{P}\_{in}^{(m)}\\mathbf{H}\_{in}^{(\\ell,m)}\\mathbf{W}\_{m}^{(\\ell+1)}+\\mathbf{P}\_{out}^{(m)}\\mathbf{\\tilde{H}}\_{out}^{(\\ell,m)}\\mathbf{W}\_{m}^{(\\ell+1)}\\Big) $$
> > >
> > >  $$= \\Big[\\mathbf{P}\_{in}^{(m)}\\mathbf{H}\_{in}\^{(\\ell,m)}+\\mathbf{P}\_{out}\^{(m)}\\mathbf{\\tilde{H}}\_{out}\^{(\\ell,m)}\\Big]\^{\\top} \\sigma^{\\prime}\\Big(\\mathbf{P}\_{in}\^{(m)}\\mathbf{H}\_{in}\^{(\\ell,m)}\\mathbf{W}\_{m}\^{(\\ell+1)}+\\mathbf{P}\_{out}\^{(m)}\\mathbf{\\tilde{H}}\_{out}\^{(\\ell,m)}\\mathbf{W}\_{m}\^{(\\ell+1)}\\Big),$$
> > >
> > > where we put the formal definition of the notations in another response due to limited characters here. The same notations can also be found in Section 3.1 in our uploaded paper. As can be seen, all variables involved in the backward propagation are within the computation graph of $\\mathcal{G}\_{m}$. In conclusion, the computation graph of $\\mathcal{G}\_{m}$ is free of those on other subgraphs, and our Eq.(34) is well-defined.

---

> ### Author Response · Authors · 2022-11-19
> **Author Response to Reviewer yVeu (Part 1)**
>
> Dear Reviewer,
>
> We appreciate that you found our paper a solid work with good theoretical and empirical results. To better answer your questions, we have provided more details and explanations **both in this rebuttal directly and in our re-uploaded paper revision**. Please refer to our response below for detail.
>
> > *“The major idea of the paper follows GNNAutoScale, and the asynchronous update is inspired by extensive research on asynchronous distributed optimization. The theoretical analyses and assumptions closely follow PipeGCN. Overall, the originality is a bit weak.”*
>
> **A1.** We respectfully disagree with the claim that our main idea follows GNNAutoScale. Rather, the three-fold, main contributions of our paper are beyond the scope of GNNAutoScale. We proposed a new, highly-parallel, and full-graph-aware distributed GNN training method; on top of this new method, we designed a novel, compute-and-storage-disaggregated training system to enable better scalability and allow distributed GNN training to potentially benefit from emerging computing paradigms and hardware; finally, we deduced new theoretical guarantees and analyses for the co-designed algorithms and systems.
>
> **(1). Methodology Novelty in Algorithm-System Co-design**: Our paper is mainly motivated from a distributed training perspective, where the proposed framework synergizes the best of both partition-based and propagation-based distributed training; GNNAutoScale provides a theoretical foundation, which exposes potential opportunities that can be harnessed by and **co-designed with new distributed training system infrastructures** to enable highly-parallel GNN training. **DIGEST goes beyond GNNAutoScale in that we built a novel distributed training framework that effectively decouples the management of state (i.e., representations) and computes (i.e., GNN training)**.
>
> **(2). System Architecture Novelty**: The disaggregated architecture of DIGEST is the result of an algorithm-system co-design as mentioned in the Methodology Novelty, and enables great properties including high scalability and low training time for large graphs, as demonstrated in our paper. More importantly, this disaggregated architecture could enable fundamental opportunities for GNN training systems to take advantage of emerging computing paradigms such as elastic serverless computing as well as emerging hardware such as Zoned Namespace SSD (ZNS) and smart programmable network hardware (SmartNIC); in our ICLR work, we have shown the promising scalability and speedup that DIGEST offers, which establishes a solid system foundation for further system-level optimizations and innovations. This demands/inspires future research along the line, which we plan to do as part of our future work.
>
> **(3). Theoretical Novelty**: All of our theoretical analyses are **tailored for a distributed training setup**, while GNNAutoScale only considers single-GPU training. We have also added a discussion in the revised paper.
>
> In our paper, specifically, we developed three theoretical analyses, i.e., gradient approximation error, synchronous convergence, and asynchronous convergence. Our analyses of gradient approximation error and asynchronous convergence are **NOT** based on PipeGCN. In fact, the PipeGCN work does not provide a  theoretical guarantee on either gradient approximation error or asynchronous convergence. Though our analyses of synchronous convergence borrow the same assumptions from PipeGCN (in fact, all assumptions in PipeGCN are widely used ones in many existing works [1,2]), the analyses are based on partition parallel (data parallel) instead of pipeline parallel training. This is also the reason why our convergence rate shown in Theorem 2 is better than that of PipeGCN (our Big O term further depends on $M^{-1/3}$, where $M$ denotes the number of machines).
>
> > *“The theoretical statements of the paper need further clarification. Assumptions being made need to be clearly clarified in the main theorem. For instance, there are lots of assumptions being made in the proof but they are never mentioned in Theorem 2 and Theorem 3. Furthermore, the constants in the Theorem need to be explained, such as $E$, $M$, $P(\eta)$ in Theorem 2 and 3."*
>
> **A2.** Thanks for your suggestions. We have added all the assumptions and explanations of the constants in our main text instead of only in our appendix. Please refer to **Pages 6-7** in our re-uploaded version for detail.
>
> **References**
>
> [1] Jianfei Chen, Jun Zhu, and Le Song. Stochastic training of graph convolutional networks with variance reduction. In International Conference on Machine Learning, pp. 942–950. PMLR, 2018.
>
> [2] Weilin Cong, Morteza Ramezani, and Mehrdad Mahdavi. On the importance of sampling in learning graph convolutional networks. arXiv preprint arXiv:2103.02696, 2021.

---

### Author Response · Authors · 2022-11-19
**General Responses From Authors**

Dear Reviewers and AC,

We thank all the reviewers for their professional and constructive comments given rather limited reviewing time for this year's ICLR. Here we address questions and comments raised by each reviewer. We briefly summarize all the revisions that we have made in the rebuttal as follows:

1. We added more discussions to better clarify the contribution of our paper beyond existing work such as GNNAutoScale and PipeGCN. Our comments can be found in our responses to the specific reviewers.

2. We added experiments for training on large graph dataset OGB-papers100m, and compared the training time per epoch with state-of-the-art GNN training frameworks.

3. We added experiments to compare DIGEST/DIGEST-A with two additional systems, PipeGCN and GNNAutoScale. Detailed results can be found in our responses to reviewers as well as in Appendix 3.4 of the revised paper.

4. We added detailed analysis and explanations about the host memory consumption and GPU memory consumption. In addition, we compared DIGEST with other baselines on GPU memory consumption on two benchmark datasets. We reported this result in the responses to reviewers and Appendix 3.2.

5. We added comparison experiments for the speedup of DIGEST under a multi-GPU, single-machine setup and a single-GPU, multi-machine setup. We reported this result in the responses to reviewers.

6. We moved some assumptions and definitions from the Appendix to the main paper to better clarify the theorems. Please refer to the Theoretical Analyses Section of the revised paper.

7. We added an additional empirical study to validate the impact of gradient approximation error and the results can be found in Appendix 3.3 of our re-uploaded version. Our results show that the error in practice quickly converges to zero and its impact is negligible.

Best,

Authors

---

> ### Author Response · Authors · 2022-12-04
> **Kind Reminder From Authors**
>
> Dear Reviewers,
>
> Thanks again for your time.
>
> The end date of Discussion Stage 2 is approaching, we hope that all your concerns have been addressed. If any concerns remain, we are happy to clarify them. Looking forward to your reply.
>
> Best regards,
> Authors

---

### Decision · Program_Chairs · 2023-01-20

**Decision:**

Reject

**Justification For Why Not Higher Score:**

The paper is present a new system for scaling GNN and it run a single experiment(in rebuttal phase) on a large graph. A more in depth experimental analysis of scalability is needed before acceptance.

**Justification For Why Not Lower Score:**

N/A

**Metareview: Summary, Strengths And Weaknesses:**

The paper introduces a new technique for distributed training for GNN. The main idea behind the paper is to combine the "partition-based" and the "propagation-based" approach to obtain a system that is scalable and accurate at the same time.

The paper presents some interesting ideas and some nice experimental results but it has few fundamental drawbacks that should be addressed before publication.

First, the experimental analysis is nice but not fully convincing. After rebuttal, the paper presents a single experiment on a large size dataset. This is not sufficient for a paper that introduces new scalable systems. Multiple large datasets should be analyzed and an in depth scalability analysis on different architecture and on graph of different size should be presented.

Second, the paper should better clarify the novelty in comparison with previous work and should improve the clarity of the theoretical results.

Overall, the paper contains some interesting ideas but in the current state it does not meet the ICLR acceptance bar.